# Ultraviolet astronomical spectrograph calibration with laser frequency combs from nanophotonic lithium niobate waveguides

Markus Ludwig[1,9], Furkan Ayhan [2,9], Tobias M. Schmidt [3,9], Thibault Wildi [1], Thibault Voumard[1], Roman Blum[4], Zhichao Ye [5], Fuchuan Lei [5], François Wildi[3], Francesco Pepe[3], Mahmoud A. Gaafar [1], Ewelina Obrzud[4], Davide Grassani[4], Olivia Hefti[4], Sylvain Karlen [4], Steve Lecomte[4], François Moreau[6], Bruno Chazelas[3], Rico Sottile[6], Victor Torres-Company [5], Victor Brasch [7], Luis G. Villanueva [2], François Bouchy[3] & Tobias Herr [1,8] ✉

Astronomical precision spectroscopy underpins searches for life beyond Earth, direct observation of the expanding Universe and constraining the potential variability of physical constants on cosmological scales. Laser frequency combs can provide the required accurate and precise calibration to the astronomical spectrographs. For cosmological studies, extending the calibration with such *astrocombs* to the ultraviolet spectral range is desirable, however, strong material dispersion and large spectral separation from the established infrared laser oscillators have made this challenging. Here, we demonstrate astronomical spectrograph calibration with an astrocomb in the ultraviolet spectral range below 400 nm. This is accomplished via chip-integrated highly nonlinear photonics in periodically-poled, nano-fabricated lithium niobate waveguides in conjunction with a robust infrared electro-optic comb generator, as well as a chip-integrated microresonator comb. These results demonstrate a viable route towards astronomical precision spectroscopy in the ultraviolet and could contribute to unlock the full potential of next-generation ground-based and future space-based instruments.

Precise astronomical spectroscopy has led to the Nobel-Prize-winning discovery of the first exo-planet via the radial velocity method[1], which relies on the precise tracking of minute Doppler shifts in the stellar spectrum caused by an orbiting planet. The detection of Earth-like planets in the habitable zone around a Sun-like star requires measuring a radial velocity change as small as 10 cm/s (relative Doppler-frequency shift of $3 \times 10^{-10}$) over the time scale of 1 year. This necessitates regular wavelength calibration of the spectrograph against a suitable calibration light source. Highly stable and accurate calibration light sources hence take on a key role in astronomical precision spectroscopy.

Calibration sources with outstanding accuracy and precision can be provided by laser frequency combs[2,3] (LFCs). Their spectra are comprised of large sets of discrete laser lines whose frequencies can be precisely known when linked to a metrological frequency standard, on

[1]Deutsches Elektronen-Synchrotron DESY, Notkestr. 85, 22607 Hamburg, Germany. [2]École Polytechnique Fédérale de Lausanne (EPFL), 1015 Lausanne, Switzerland. [3]Observatoire de Genève, Département d'Astronomie, Université de Genève, Chemin Pegasi 51b, 1290 Versoix, Switzerland. [4]Swiss Center for Electronics and Microtechnology (CSEM), 2000 Neuchâtel, Switzerland. [5]Department of Microtechnology and Nanoscience, Chalmers University of Technology, 41296 Gothenburg, Sweden. [6]Observatoire de Haute-Provence, CNRS, Université d'Aix-Marseille, 04870 Saint-Michel-l'Observatoire, France. [7]Q.ANT GmbH, Handwerkstraße 29, 70565 Stuttgart, Germany. [8]Physics Department, Universität Hamburg UHH, Luruper Chaussee 149, 22607 Hamburg, Germany. [9]These authors contributed equally: Markus Ludwig, Furkan Ayhan, Tobias M. Schmidt. ✉e-mail: tobias.herr@desy.de

a level that exceeds the astronomical requirements. Provided their lines can be separated by the spectrograph, LFCs can serve as exquisite wavelength calibrators[4,5] and are then also referred to as astrocombs (Fig. 1a). Recent years have seen significant progress in developing astrocombs[6–10] and extending their coverage to visible (VIS) wavelengths, where the emission of Sun-like stars peaks, and to near-infrared wavelength, where relatively cold stars can accommodate rocky planets in tight orbits.

Extending astrocombs further towards the atmospheric cutoff and into the ultraviolet (UV) regime would not only be relevant to exoplanet science, but highly desirable for fundamental physics and precision cosmology (Fig. 1b). Indeed these are also major science drivers for the upcoming ANDES high-resolution spectrograph at the extremely large telescope (ELT)[11]. For instance, by accurately determining the wavelengths of narrow metal absorption features in the spectra of quasi-stellar objects (QSOs), one can infer the value of the fine-structure constant, $\alpha$, at the place and time of absorption, i.e. billions of years ago. This allows us to test whether fundamental physical constants might have varied on cosmological scales[12,13]. Another

exciting, but extremely demanding science case is the direct observation of cosmic expansion in real time, often referred to as the Sandage test[14–16]. This requires detecting a drift of H I Lyα forest absorption features in QSO spectra by just a few cm/s (relative frequency shift of $3 \times 10^{-11}$) over several decades. As lined out in ref. 17, observations and corresponding wavelength calibrations in the UV are needed to access low redshifts, $z$, in particular, the important zero-crossing of the drift at $z = 2$, corresponding to 365 nm (Fig. 1c). Achieving such an extreme level of wavelength stability will require a community effort over the next decade to develop improved spectrographs, data analysis methods, and in particular LFC-based calibration sources[18]. Furthermore, it has recently become clear that a detailed modeling of the instrumental line-spread function is crucial to further improve the accuracy and stability of spectroscopic observations[17,19,20]. This, however, requires a broadband calibration source with a spectrum of narrow lines, such as those of an LFC.

Astrocombs can be derived from high-repetition-rate ($f_{rep} > 10$ GHz) laser sources, such as line-filtered mode-locked lasers[6–8,21], electro-optic (EO) comb generators[22,23] and chip-integrated microresonators[24–28]. Their spectra consist of equidistant optical laser frequencies $\nu_n = \nu_0 + n f_{rep}$, spaced by the laser's pulse repetition rate $f_{rep}$ and with an offset frequency $\nu_0$, which we here define to be the central frequency of the laser ($n$ is an integer comb line index). To transfer the comb spectra to the desired wavelength range and to increase their spectral coverage, second- ($\chi^{(2)}$) and third-order ($\chi^{(3)}$) nonlinear optical effects are utilized. However, the high-repetition rates of >10 GHz, that are necessary for the comb lines to be separated by the spectrograph, imply low pulse energies and hamper nonlinear optical effects. In addition, and especially at shorter wavelengths, strong material dispersion complicates the efficient nonlinear energy transfer.

So far, short-wavelength blue and green astrocombs have been achieved via second-harmonic generation of infrared mode-locked lasers[21,29,30] and of EO combs[31] in bulk $\chi^{(2)}$-nonlinear crystals. Spectral broadening in $\chi^{(3)}$-nonlinear photonic crystal fiber resulted in astrocombs spanning from infrared (IR) to VIS wavelength[32–34], some extending to below 450 nm. In addition to these approaches, micro- and nanophotonic waveguides made of highly nonlinear materials can further increase the efficiency in a compact setup. In the IR, $\chi^{(3)}$-nonlinear silicon nitride (Si$_3$N$_4$) waveguides have led to broadband astrocombs[35], and via third-harmonic generation, a 10 GHz IR comb has been transferred to VIS wavelengths[36]. Strong nonlinear effects in waveguides, including harmonic frequency generation, may also be obtained via the quadratic $\chi^{(2)}$-nonlinearity. At low pulse repetition rates, spectra across multiple optical octaves have been obtained in large-mode area (~100 μm$^2$) periodically poled lithium niobate (LiNbO$_3$) waveguides with mm-long propagation distances[37–40]. Notably, periodically poled LiNbO$_3$ waveguides were utilized with a 30 GHz repetition rate line-filtered erbium-doped mode-locked laser[41], creating harmonic spectra in VIS and UV domains for astronomical spectrograph calibration. In addition, recent work leveraged a combination of second-harmonic and sum-frequency generation based on a titanium-doped sapphire laser to obtain, in conjunction with line-filtering, a continuous green to ultraviolet 30 GHz astrocomb[42]. In nanophotonic waveguides from thin-film LiNbO$_3$[43–48], owing to their sub-μm$^2$ modal confinement, highly-efficient UV spectral generation has been reported[49,50,50–59], including at 10 GHz repetition rate[59]. However, astronomical spectrograph calibration in the UV spectral range has so far not been demonstrated.

Here, we demonstrate astronomical spectrograph calibration with laser frequency combs in the UV below 400 nm. Specifically, we design chip-integrated nano-structured LiNbO$_3$ waveguides with sub-μm$^2$ mode cross-section and a tailored poling pattern to enable efficient UV astrocomb generation via cascaded harmonic generation from a robust telecommunication-wavelength 18 GHz EO comb, as well as, a

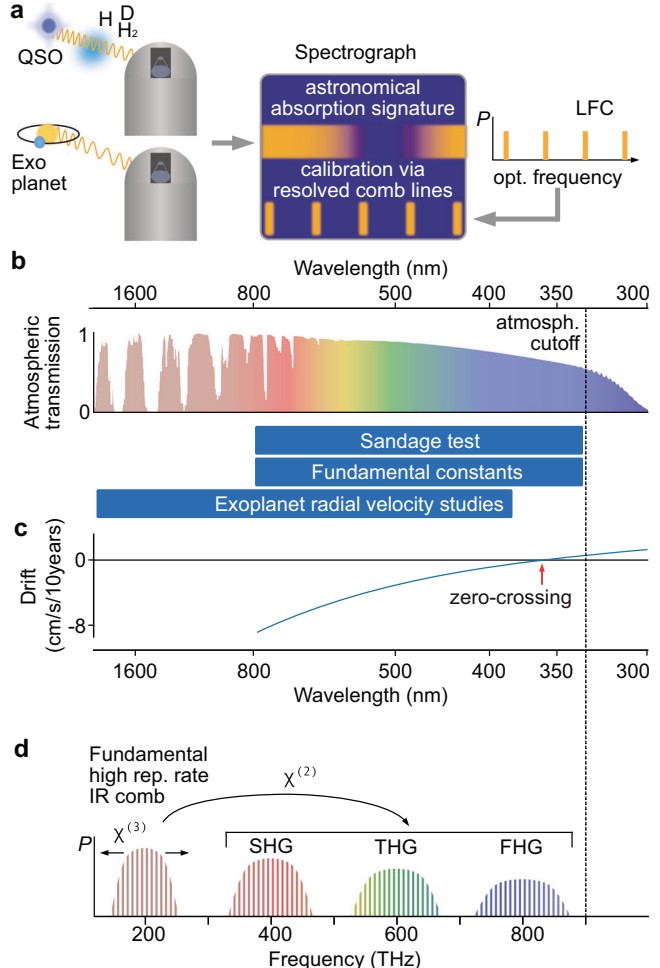

**Fig. 1 | Ultraviolet astronomical precision spectroscopy. a** Precision spectroscopy of astronomical objects enabled by laser frequency combs (LFC) providing absolutely and precisely known laser frequencies, cf. main text. QSO: quasi-stellar object, H, H$_2$: hydrogen, D: deuterium, P: optical power. **b** Atmospheric transmission (high altitude, dry), spectral ranges for different astrophysical studies. **c** Expected redshift drift signal for the standard ΛCDM cosmology[89] in units of cm/s per decade (Sandage test) as a function of wavelength, corresponding to the redshifted H I Lyα transition. **d** Concept of broadening and transferring a high-repetition-rate astrocomb from infrared (IR) to visible (VIS) and ultraviolet (UV) wavelengths based on optical $\chi^{(2)}$- and $\chi^{(3)}$-nonlinearities.

25 GHz chip-based microresonator comb. The generated astrocombs are tested in March 2023 on the high-resolution echelle spectrograph SOPHIE[60] at the Observatoire de Haute Provence (OHP, France), achieving radial-velocity precision in the range of 1 to 2 m/s.

## Results

We base our astrocombs on a robust, telecommunication-grade EO comb with a pulse repetition rate of 18 GHz, as well as, a microresonator comb with a repetition rate of 25 GHz. Both are derived from an IR continuous-wave (CW) laser of frequency $\nu_0 \approx 192$ THz (wavelength ca. 1.56 μm) and operate intrinsically at a high-repetition rate, so that filtering of unwanted comb lines is not required. To transfer the IR comb to the UV (and VIS), we utilize chip-based LiNbO₃ nanophotonic waveguides, for cascaded harmonic generation as illustrated in Fig. 1d. The possible comb frequencies are

$$\nu_{m,n} = m\nu_0 + nf_{\text{rep}} \qquad (1)$$

where $m$ is the harmonic order. In general, each harmonic $m$ of the comb spectrum has a different offset frequency $m\nu_0$. While this can facilitate the detection of the comb's offset frequency[51,55,57,59], care must be taken to avoid inconsistent frequency combs from overlapping harmonics with different offsets. Alternatively, stabilizing $\nu_0$ to a multiple of $f_{\text{rep}}$ could address this point. Practically, this could be achieved, for instance, by locking the beatnote between adjacent harmonics to zero[61], or via an auxiliary low-repetition rate frequency comb[62].

A key difference between broadband electro-optic combs obtained via conventional self-phase modulation $\chi^{(3)}$-based supercontinua and spectra generated via $\chi^{(2)}$-based harmonics, lies in the phase noise properties of their comb lines. In $\chi^{(3)}$-based spectra, the phase noise, impacting linewidth and line contrast, grows quadratically with the frequency distance from the fundamental pump laser. In contrast, in $\chi^{(2)}$-based harmonic spectra the phase noise (approximately) depends on the frequency distance from its corresponding harmonic's center frequency (Supplementary Information SI, Section 1 and Supplementary Fig. 1). This property, along with the efficiency of the nanophotonic waveguides, permits us to directly generate comb lines in the ultraviolet from a telecom-wavelength electro-optic source, within a certain bandwidth around the harmonic's center frequency; future extension of this bandwidth will, as we discuss in detail in the Section 1 in the SI and show in the Supplementary Figs. 2, 3, require the addition of a noise filtering cavity, which has been established for broadband $\chi^{(3)}$-based electro-optic combs[63], but is not implemented in this initial proof-of-concept demonstration.

### Waveguide design

An illustration of a nano-structured LiNbO₃ waveguide and the strong mode confinement to sub-μm² cross-section for efficient nonlinear conversion is shown in Fig. 2a. The waveguides are fabricated from a commercially available undoped lithium niobate-on-insulator (LNOI) substrate with an 800-nm-thick LiNbO₃ layer via electron-beam lithography and reactive-ion etching, resulting in a sidewall angle of ~75° (Fig. 2c, Methods, SI Section 2 and Supplementary Figs. 4, 5). The waveguides taper to a width of 2500 nm at the chip facet, for input and output coupling. In the first design step, considering the fundamental transverse electric field polarization (TE) waveguide mode, we define the waveguide geometry, which in nanophotonic waveguides enables engineering of the waveguide's group velocity dispersion (GVD), and additional $\chi^{(3)}$-nonlinear spectral broadening of the input IR comb. In unpoled waveguides, top widths in the range of 800 to 1200 nm result in anomalous GVD and the most significant broadening. Spectral broadening is also possible in the normal GVD regime, typically resulting in less broad, but more uniform spectra.

In a second design step, we define the waveguide's poling pattern to transfer the broadened fundamental spectrum to shorter wavelengths. This can be achieved efficiently, when a fixed phase relation between the involved waves is maintained (phase matching). However, due to chromatic material dispersion, this cannot easily be fulfilled across the entire spectral interval of interest (regardless of the choice of waveguide geometry and GVD at the fundamental wavelength). As an alternative to natural phase matching, in ferroelectric crystals such as LiNbO₃, quasi-phase matching (QPM) is possible. Here, the sign of the $\chi^{(2)}$-nonlinearity is periodically flipped within segments of length $\Lambda$ (poling period), via poling of the crystal's domains (Methods) to compensate the increasing phase-mismatch between the waves. For sum-frequency generation processes (which underlie harmonic generation), $\Lambda$ is

$$\Lambda = 2\pi / |k_1 + k_2 - k_3| \qquad (2)$$

where $k_1$ and $k_2$ are the input propagation constants, and $k_3$ is the output propagation constant in the waveguide. To achieve broadband QPM, $\Lambda$ may be chirped (i.e., varied along the waveguide). To design $\Lambda(z)$ ($z$ is the spatial coordinate in the propagation direction), we numerically compute the frequency $\omega$ dependent propagation constants $k(\omega)$ of the fundamental waveguide mode for waveguide top widths of 800, 1000, and 1200 nm. Based on Eq. (2), we derive the required poling period $\Lambda$ for a number of sum-frequency processes that would result in second-harmonic generation (SHG = fundamental + fundamental), third-harmonic generation (THG = SHG + fundamental), and fourth-harmonic generation (FHG = SHG + SHG or THG + fundamental). These periods $\Lambda$ are shown in Fig. 2b, as a function of the fundamental comb's wavelengths.

Prior to astrocomb generation, we tested different waveguide designs with a low-repetition rate (100 MHz), 80 fs mode-locked laser operating at a center wavelength of 1560 nm. The pulses are coupled to the chip via a high-numerical aperture lens (NA = 0.7) lens and the spectra are collected using reflective collimation optics and a fluoride multimode fiber that connects to a grating-based optical spectrum analyzer (Yokogawa AQ6374). Utilizing a lensed fiber with a calibrated coupling efficiency as a temporary output coupler, we determine the input coupling efficiency of the high-NA lens to be 13 ± 2%. To illustrate the impact of the poling structure, we compare in Fig. 2d a waveguide poled for broadband SHG ($\Lambda$ weakly chirped as indicated) with an unpoled waveguide (Fig. 2e), demonstrating markedly different behaviors. The poled waveguide creates a broadband SHG at ~400 THz, whereas the unpoled waveguide only creates a narrow SHG. These observations agree with numerical simulation via pyChi[64] (shown below the experimental spectra). Note that the poled waveguide also exhibits a more substantial broadening of the input spectrum (and its harmonics), which can be explained through the contributions of cascaded $\chi^{(2)}$-processes to the effective $\chi^{(3)}$-nonlinearity. In a more advanced design (Fig. 2f) a short SHG section (supporting also broadening) is followed by a longer and strongly chirped section designed to enable FHG. Indeed, pronounced FHG is visible, although, in contrast to the simulation, it is not stronger than the THG signal. We attribute this to the very small values of $\Lambda < 2$ μm, which are challenging to fabricate with high fidelity. To overcome the fabrication challenge for very small $\Lambda$, we also consider higher-order QPM (Fig. 2b). In higher-order QPM, the poling period is an integer multiple $q$ of the fundamental poling period $\Lambda$[65], which relaxes fabrication requirements at the cost of reduced efficiency. In our design for symmetric poling (equal length for both crystal orientations), $q$ is an odd-integer (here: $q = 3$). In addition, unintentional phase-matching may occur involving higher-order waveguide modes and coupling between orthogonal polarizations. A possible design is shown in Fig. 2g, where, after a short length of SHG poling, $\Lambda$ ranges from 2 to 7 μm. This choice of $\Lambda$ respects the fabrication limit and provides QPM

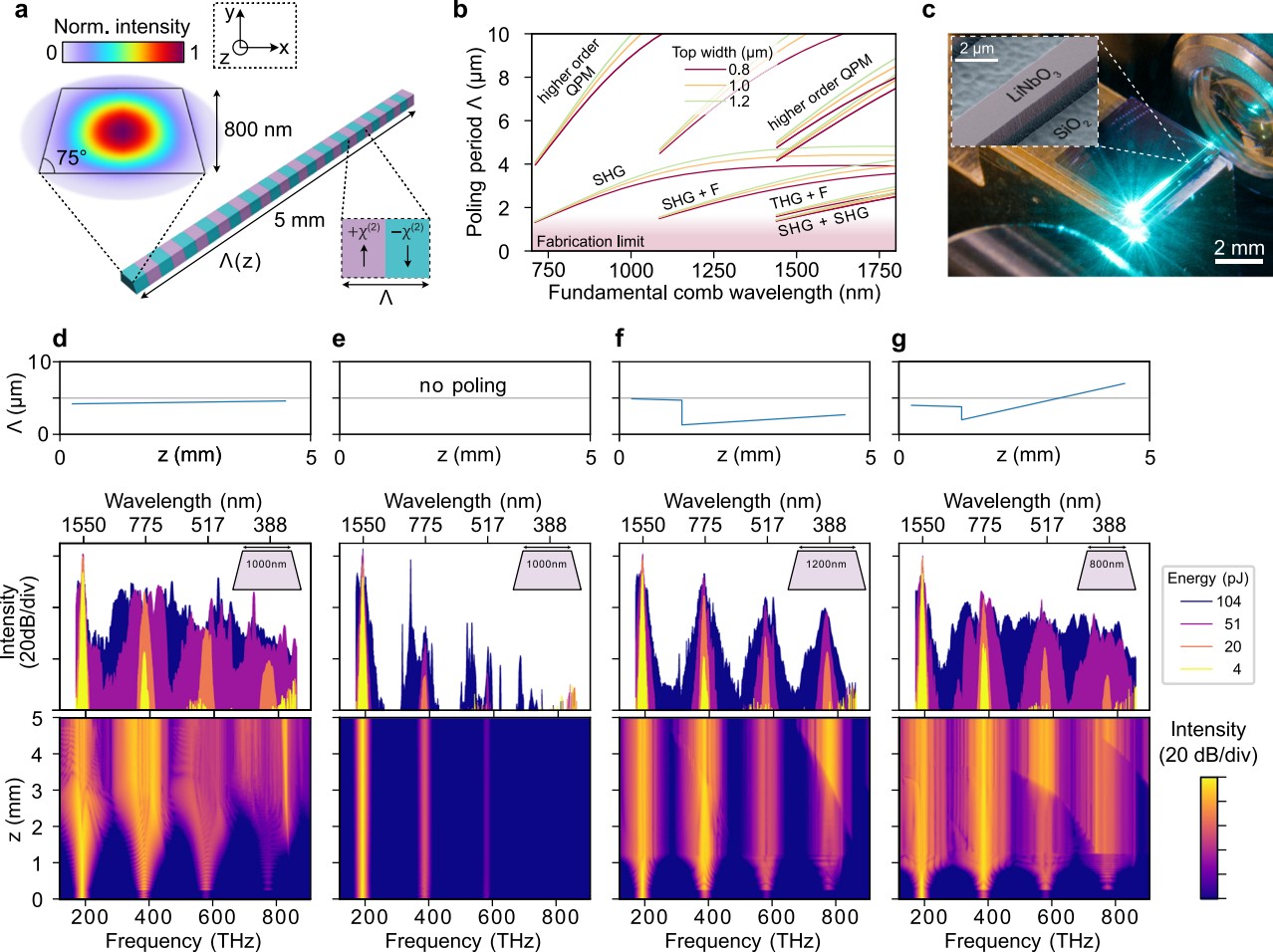

**Fig. 2 | Nanophotonic lithium niobate waveguides. a** Geometry of a nanophotonic lithium niobate (LiNbO$_3$) waveguide, showing the optical mode profile for the fundamental transverse electric field polarization (TE) mode and a possible poling pattern with spatially variable poling period $\Lambda$. The extraordinary crystal axis is oriented along the x-direction to access the highest electro-optic tensor element of LiNbO$_3$. **b** Poling periods $\Lambda$ required for quasi-phase-matching of various nonlinear optical processes for waveguides of different (top) width as a function of the fundamental comb's wavelength. SHG: second-harmonic generation, SHG+F: sum-frequency generation of second-harmonic and fundamental comb, THG+F: sum-frequency generation of third-harmonic and fundamental comb, SHG+SHG: sum-frequency generation of the second-harmonic. Respective higher-order phase matching with three-fold period $\Lambda$ is indicated (the current fabrication limit only allows for $\Lambda > 2\,\mu m$). **c** Photograph of a waveguide in operation and scanning electron microscope image of the LiNbO$_3$ on silica (SiO$_2$). **d–g** Examples of waveguide designs showing poling pattern (top), experimentally generated spectra for different input pulse energies provided by a 100 MHz, 80 fs mode-locked laser with a central wavelength of 1560 nm (waveguide cross-section are shown as insets and on chip-pulse energy is indicated by the color code) and *pyChi*[64] simulation results (bottom) for a pulse energy of 50 pJ. The spurious spikes observed in the traces for 4 pJ pulse energy are manifestations of the noise floor of the optical spectrum analyzer (Yokogawa AQ6374).

for most processes (Fig. 2b). The generated harmonic spectrum extends across 600 THz (ca. 350 to 1000 nm) within ~20 dB of dynamic range, demonstrating, in agreement with previous work[59], the potential of LiNbO$_3$ waveguides for generating gap-free spectra in the VIS and UV domains.

**Spectrograph calibration**

With the principle of efficient harmonic generation validated, we build a dedicated ultraviolet astrocomb and use it for spectrograph calibration under realistic conditions at the high-resolution echelle spectrograph SOPHIE[60] at the Observatoire de Haute Provence (OHP, France). With the help of an Echelle-grating and a prism, the spectrograph cross disperses the input spectrum on a two-dimensional charged coupled device (CCD) detector array so that the spectrum is arranged in a vertically separated set of nearly horizontal lines (*Echelle orders*), where each line represents a distinct wavelength interval. The SOPHIE spectrograph, covers the wavelengths $\lambda$ from 387 to 708 nm with a resolving power of $R = \frac{\lambda}{\Delta\lambda} \approx 75'000$ (i.e., frequency resolution

of approximately 10 GHz at 400 nm wavelength) and its echelle-grating is enclosed in a pressure vessel to reduce externally induced drifts. Traditionally, wavelength calibration can be obtained via thorium-argon hollow-cathode lamps (ThAr) and a passively-stabilized Fabry-Pérot etalon (FP)[66,67], permitting calibration on the 1 to 2 m s$^{-1}$ level[68].

As an initial IR comb generator, we utilize an 18 GHz EO comb, similar to ref. 62 (Fig. 3a). A CW laser of wavelength 1556.2 nm is stabilized to an optical resonance frequency in rubidium (Rb), providing an absolute frequency anchor for the IR EO comb (Methods). Comb generation proceeds by sending the CW laser through two electro-optic phase- and one electro-optic intensity modulator, creating an initial IR comb spanning ~0.6 THz. The modulation frequency of 18 GHz is provided by a low-noise microwave synthesizer, referenced to a miniature Rb atomic frequency source; over the relevant time scales of our experiments, a relative frequency stability better than 10$^{-11}$ is achieved (Methods). In a waveshaper, a polynomial spectral phase up to third order is imprinted on the EO comb spectrum, to

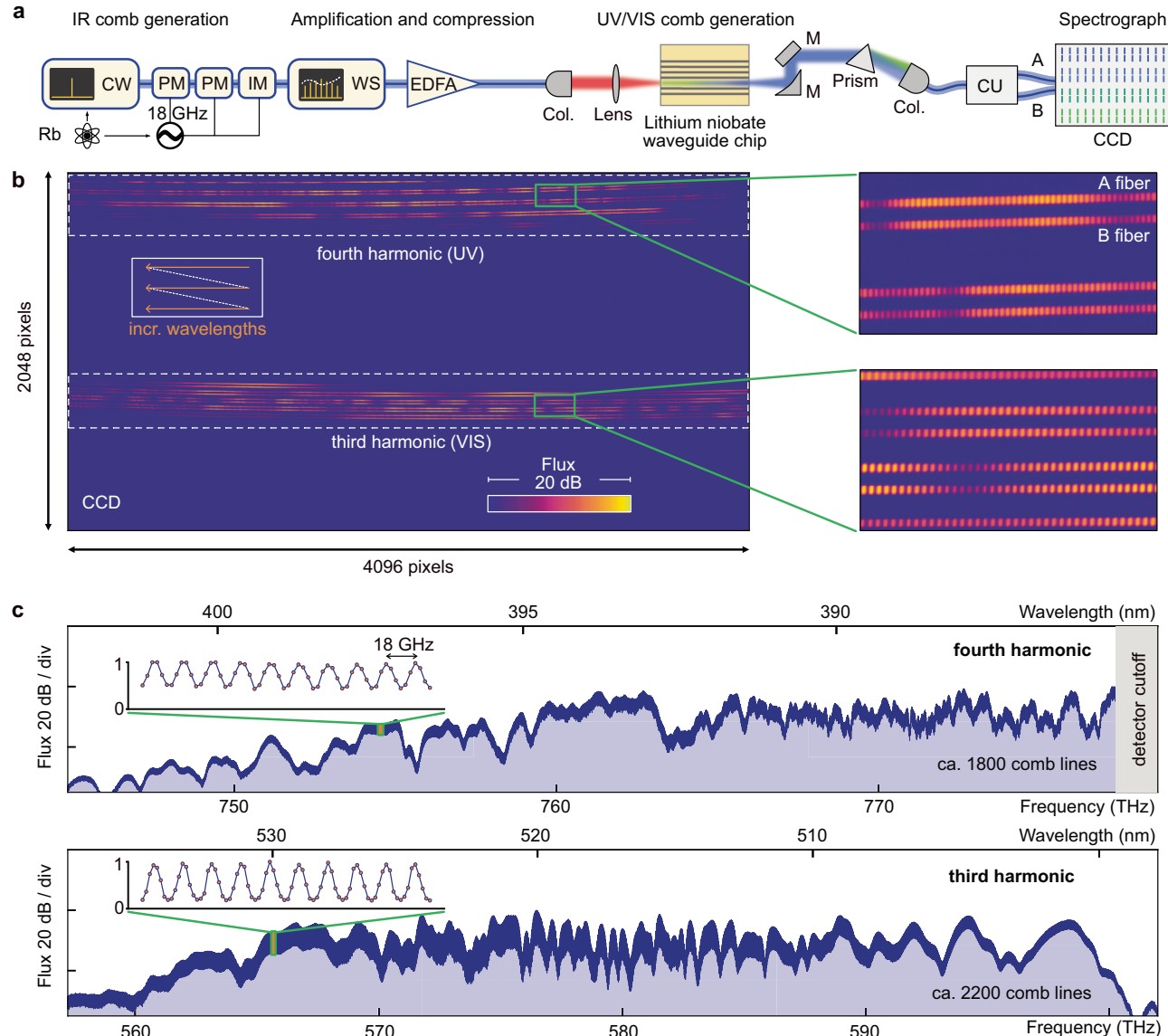

**Fig. 3 | Ultraviolet astrocombs. a** Setup for ultraviolet (UV) astrocomb generation. Rb rubidium-based atomic references (main text, Methods); CW continuous-wave laser, PM phase modulator, IM intensity modulator, WS waveshaper, EDFA erbium-doped fiber amplifier, Col. fiber-to-free-space collimator, M mirror, CU calibration unit splitting the comb equally into A and B fibers, CCD SOPHIE charged coupled device detector. All fibers following the lithium niobate chip and feeding the spectrograph are multimode fibers. **b** Composite CCD detector image, obtained with two different settings of the flux-balancing prism, showing fourth- and third- harmonic in the UV and visible (VIS) wavelength ranges. The inset indicates how wavelengths increase along the individual cross-dispersed echelle orders. The magnified CCD sub-images show the distinguishable but slightly overlapping frequency comb lines on the detector, originating from the two spectrograph fibers, which are both fed by the same comb. **c** Extracted 1-dimensional comb spectra of both harmonics; the insets show a magnified (linear scale) portion of the spectrum and the 18 GHz-spaced comb lines. The UV detector cutoff is marked.

ensure short pulses after the amplification in an erbium-doped fiber amplifier (EDFA) and an additional stretch of optical fiber for nonlinear pulse compression. In this way, an 18 GHz femtosecond pulse train with 3 W of average power can be generated. The pulses are out-coupled to free space and coupled to the chip via a high-numerical aperture (NA = 0.7) lens (estimated on-chip-pulse energy ~20 pJ). Despite the high average input power levels, we did not observe any waveguide damage.

For comb generation, we choose the waveguide in Fig. 2f, as it exhibits a strong UV signal and a clear separation between the harmonics. We arrange the waveshaper and the optical fiber after the amplifier such that the driving pulses are of ca. 200 fs duration, maintaining the separation between the harmonics, so that the emergence of inconsistent frequency combs, as described above, can be excluded. As we show in the SI, Section 3 and Supplementary Fig. 6, shorter pulses can enable the generation of broadband and merging harmonics, which opens opportunities for future work. The generated spectra are similar to those generated at a lower repetition rate (Fig. 2d–g). The chip's output, including fundamental and higher-order waveguide modes, is collimated using a parabolic mirror and sent through a tunable prism for coarse wavelength filtering and balancing of flux levels as needed, before being coupled to a multimode fiber. The light is sent through a mode-scrambler (Methods) and then routed to an auxiliary input port in the spectrograph's calibration unit, which marks the interface between our LFC setup and the existing telescope and spectrograph infrastructure at OHP[60]. The calibration unit splits the comb light equally into two optical fibers (A/B fiber), which are placed next to each other on the spectrograph entrance slit, each

creating an equivalent cross-dispersed spectrum on the two-dimensional detector. Figure 3b shows a comb spectrum as recorded on the CCD detector of the SOPHIE spectrograph. The comb structure of the LFC spectrum is clearly visible along the individual spectral orders of the cross-dispersed echelle spectrograph for both A and B fibers. Data extraction and reduction of the LFC exposures is performed using a custom code initially developed for the ESPRESSO spectrograph[69]. It extracts the spectral information from the two-dimensional detector frame using a variant of the flat-relative optimal extraction algorithm[70], provides default wavelength calibration based on a combined ThAr/FP wavelength solution[69,71], and combines the data from the individual spectral orders to a single one-dimensional spectrum (Fig. 3c) and thus makes it accessible to subsequent analysis. The resolving power of the spectrograph is not sufficient to fully separate the comb lines. In particular, in the UV part of the spectrum, the wings of the spectrograph's line-spread functions for adjacent comb lines overlap and imply a reduced contrast. Nevertheless, the individual comb lines are clearly discernible.

To test the stability of the LFC-based wavelength calibration, we obtained over several hours repeated LFC exposures interleaved with exposures of the FP etalon. The two types of exposures allow us to measure the spectrograph drift in independent ways, once relative to the LFC and once relative to the FP. After data reduction as described above, all LFC lines in the spectra are fitted individually to determine their positions on the detector (Methods). For the FP exposures, the gradient method[72] is used to non-parametrically detect shifts of the spectra on the detector. As shown in Fig. 4a, we obtain separate drift measurements from the third- and fourth-harmonic and, in addition, one drift value from the FP, where we only use the FP lines that overlap spectrally with the third-harmonic (the FP calibration does not cover the UV). All three measurements agree well (within the

amplitude of their scatter) and detect a systematic spectrograph drift of ca. −2 m/s over 4 h. The agreement between the third-harmonic and the FP validates the LFC, and the agreement between the fourth-harmonic with the two other measurements extends the validation to the UV regime. The slightly higher noise in the LFC data, i.e., remaining differences between third- and fourth-harmonic and scatter around the global trend, compared to the FP drift measurement, can probably be attributed to a combination of effects, e.g., intrinsic systematics of the spectrograph like detector stitching issues[73] or charge transfer inefficiency (CTI)[74] combined with a change in the flux levels of the LFC lines. Moreover, the line spacing of our comb (18 GHz), which is limited by our specific technical components, is less than half the FSR of the FP calibrator (39 GHz). As a result, comb line fitting (Methods) is more challenging, and a slightly higher scatter can be expected, when compared to the FP-based calibration. Implementing a comb with larger spacing or higher resolution spectrographs can address this challenge. The fact that, nevertheless, the scatter in the FP and comb-based calibrations are comparable, indicates that the obtained precision is here limited by the spectrograph, not by the comb (indeed, the scatter is consistent with the design goals of the spectrograph[60]).

To test the accuracy of our LFC-based calibration, we compare it to the established laboratory wavelengths of the ThAr hollow-cathode lamp. For this, two exposures, one containing the LFC and the other the ThAr spectrum, are obtained in quick succession, which ensures that no relevant spectrograph drift happens between the two exposures. In the extracted LFC spectrum, individual lines are fitted (see above and Methods) and, together with the knowledge about their true, intrinsic frequencies (Eq. (1)), a purely LFC-based wavelength solution, i.e., a relation between CCD pixel position and wavelength, is derived for the spectral region covered by the comb. This LFC

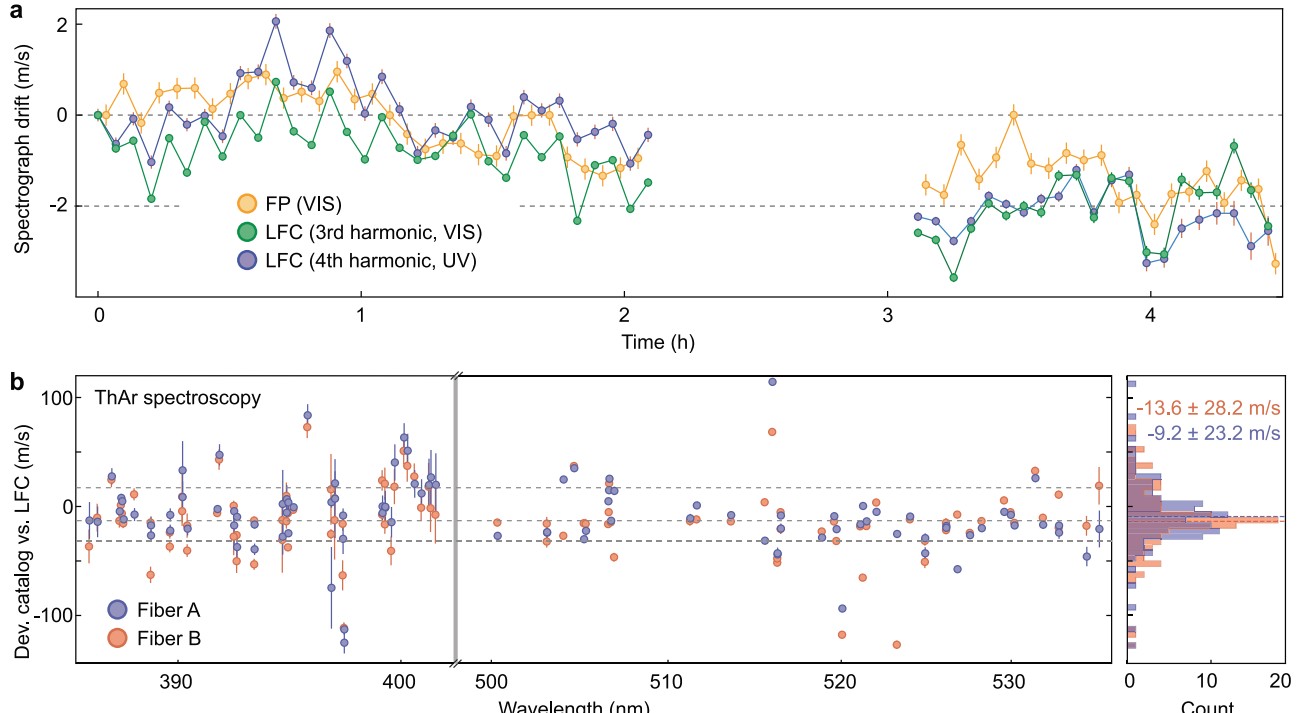

**Fig. 4 | Spectrograph calibration. a** Spectrograph drift measurement based on the Fabry-Pérot (FP) etalon in the visible (VIS) wavelength and the laser frequency comb (LFC) in VIS and ultraviolet (UV) wavelength range. The FP calibration data has been restricted to its spectral overlap with the VIS LFC; Error bars indicate formal uncertainties based on propagation of photon- and read-out noise. **b** Absolute spectroscopy of thorium emission lines by the LFC-calibrated spectrograph. The

deviation of the literature values from our LFC-calibrated measurement is shown as a function of wavelength and separately for A and B fiber. Error bars indicate formal fitting uncertainties based on a propagation of photon- and read-out noise; dashed lines indicate 16th, 50th, and 84th percentiles. Right panel: histogram of the deviations, separated for Fiber A and B. Mean and standard deviation are indicated.

wavelength solution is then applied to the extracted ThAr spectrum and, after fitting of line centroids, LFC-calibrated wavelengths are assigned to the thorium lines. These are then compared to the laboratory wavelengths[75] and the deviations are shown in Fig. 4b. The histogram of all deviations shows a global offset of −9 to −14 m/s with a standard deviation of >20 m/s, which is consistent with zero offset, and also with an equivalent comparison performed with the highly accurate ESPRESSO spectrograph[69]. We note that the large uncertainty (the standard deviation in the histogram) is not limited by the comb calibrator but due to the low number of only ~100 ThAr lines that are available for comparison. The scatter of individual line deviations (and the difference between Fibers A and B), which significantly exceeds the formal uncertainties, has also been observed for other spectrographs (e.g., ESPRESSO[69]) and seems to be a common systematic, probably related to an imperfect determination of the instrumental line-spread function. We confirm that this scatter is identical when comparing thorium lines to LFC or ThAr/FP wavelength solutions[69,71]. Therefore, we conclude that the LFC wavelength calibration is accurate to the level testable with SOPHIE. Both measurements combined validate on the proof-of-concept level that the generated astrocomb can provide precise and absolute calibration even at UV wavelengths.

## Microresonator-based UV astrocombs

Finally, we explore photonic chip-integrated microresonators (microcombs[24–26]) as the fundamental IR comb generator. While their current level of maturity is lower than that of EO comb generators or mode-locked lasers, microcombs can provide compact and energy efficient[76] high-repetition rate, low-noise femtosecond pulses[77], and their potential for microresonator astrocombs has already been demonstrated in the IR spectral range[27,28]. Combined with highly-efficient frequency conversion in LiNbO$_3$ waveguides, potentially in a resonant configuration[78–80], they could open new opportunities for space-based calibrators, where energy consumption, size, and robust integration are key. Such space-based systems could calibrate ground-based observatories through the atmosphere, or provide calibration to space-based observatories, which would entirely avoid limiting atmospheric effects, such as absorption or turbulence. Therefore, as a proof-of-concept, we replace the electro-optic comb generator with a silicon nitride microresonator[81,82] (Fig. 5c, photo) operating at $f_{rep} = 25$ GHz (Methods). A portion of the generated comb spectrum is bandpass-filtered and pre-amplified in an erbium-doped fiber amplifier (Fig. 5a). Figure 5b shows the IR comb spectrum generated by the microresonator as well as the filtered and amplified spectrum. The remainder of the setup is the same as described before for the EO astrocomb (Fig. 3a). While we do not stabilize the microcomb spectra, we observe broadband harmonics of distinct comb lines, similar to the EO comb configuration (Fig. 5c), illustrating the potential of implementing all nonlinear optical components via highly-efficient photonic integrated circuits.

Although lithium niobate is known to show photo-induced damage by visible and ultraviolet light, we did not observe any signs of degradation in our experiments and attribute this to the relatively low flux levels in these wavelength domains (The power in the harmonics and corresponding conversion efficiencies are presented in the SI, Section 4 and Supplementary Fig. 7). In a dedicated experiment, described in Section 5 of the SI and Supplementary Fig. 8, we transmitted light of a 405 nm laser diode through the waveguide and did also not observe any degradation. Zirconium-doping of the lithium niobate can further increase the damage threshold for visible and ultraviolet light[83].

## Discussion

In this work, we demonstrate astronomical spectrograph calibration in the UV wavelength range with a laser frequency comb. This

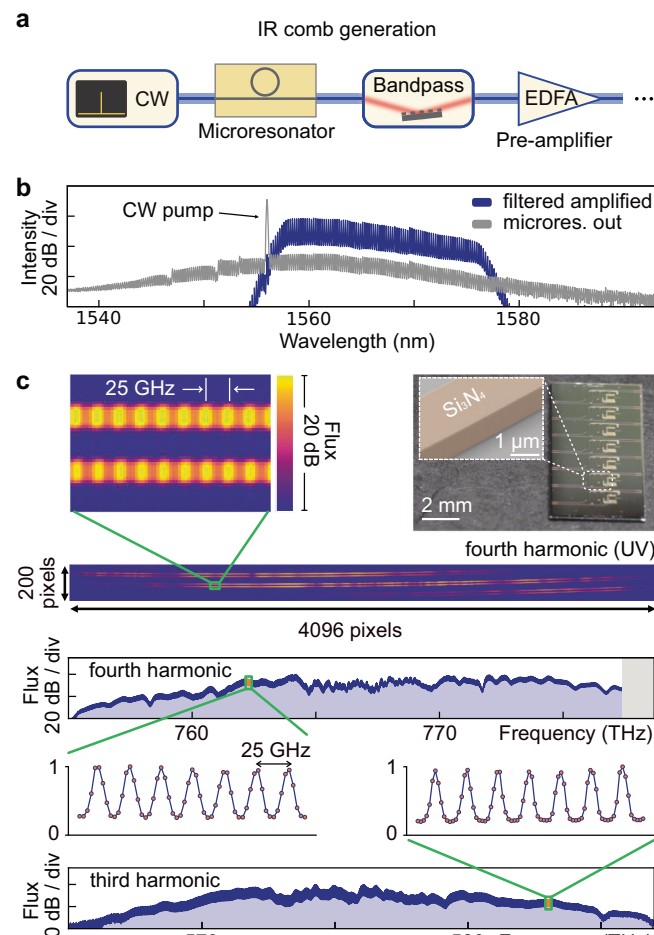

**Fig. 5 | Microresonator-based ultraviolet astrocombs. a** Modified setup for microresonator-based ultraviolet (UV) astrocomb generation. **b** Microresonator comb output spectrum (gray); bandpass-filtered and pre-amplified microresonator-based comb spectrum. **c** Microresonator-based fourth-harmonic and visible (VIS) third-harmonic comb spectra, following the scheme in Fig. 3. The photo shows the photonic chip with the microresonator (dashed-line box) and the inset shows a scanning electron microscope image of the Si$_3$N$_4$ resonator waveguide.

demonstration is particularly relevant to the emerging field of optical precision cosmology with next-generation instruments and telescopes. Our astrocomb's architecture is based on periodically poled lithium niobate nanophotonic waveguides tailored to provide efficient frequency conversion of a robust, alignment-free 18 GHz electro-optic frequency comb. Moreover, we show the compatibility of the approach with photonic chip-integrated microcombs, creating opportunities for future space-based systems. This nonlinear transfer of a microcomb to VIS and UV wavelengths also adds new operating domains to the repertoire of these sources. The approach demonstrated here can be utilized to extend calibration beyond established wavelength ranges towards shorter wavelengths. Future work may pursue wider harmonic spectra (SI, Section 5), where however, consistency of the generated spectra via suitable control of the fundamental comb's offset-frequency (as described above) and noise suppression as previously demonstrated for electro-optic combs[63] would need to be implemented. With improved coupling to the waveguide[47,84] this could enable gap-free VIS and UV operation based on robust telecommunication-wavelength IR lasers. Similar results may be possible also in other waveguide materials such as aluminum nitride (AlN), where broadband phase matching may be achieved through chirped geometries[85]. In addition to the development of the

comb source itself, spectral flattening[86] will be needed to ensure optimal exposure of the CCD and to avoid systematic effects that can arise from strong modulation of the comb's spectral envelope. In sum, the demonstrated approach opens a viable route towards astronomical precision spectroscopy in the ultraviolet based on robust infrared lasers and may eventually contribute to major scientific discoveries in cosmology and astronomy.

## Methods

### Waveguide and microresonator fabrication

Lithium niobate waveguides for UV comb generation are fabricated from an 800-nm-thick x-cut $LiNbO_3$ layer on $3\,\mu m$ $SiO_2$ and bulk Si substrate (NANOLN). First, chromium (Cr) electrodes for periodic poling are fabricated using electron-beam lithography and lift-off. Ferroelectric domain inversion is performed by applying a high-voltage waveform across the electrodes, adapted from ref. [87] (SI, Fig. 5). After poling, the electrodes are removed via chemical etching. Waveguides are patterned using electron-beam lithography on a Cr hard mask. Cr is etched using argon (Ar) ion-beam etching and subsequently, the lithium niobate layer is fully etched (no slab remaining) via reactive-ion etching with fluorine chemistry. The remaining Cr hard mask is later removed with chemical etching. Lastly, a $3$-$\mu m$-thick $SiO_2$ cladding layer is deposited using chemical vapor deposition and waveguide facets are defined by deep etching to ensure low-loss coupling to the waveguides. More details regarding the fabrication of lithium niobate waveguides can be found in SI, Section 2.

The $Si_3N_4$ microresonator is designed with a finger shape[82] and fabricated via the subtractive processing method as described in ref. [81]. The height and the width of the SiN waveguide are 740 nm and 1800 nm respectively, which result in a group velocity dispersion coefficient of $\beta_2 = -73\,ps^2/km$ for the fundamental TE mode at the pump wavelength. The total linewidth of the critically coupled resonator is 25 MHz. Lensed fibers are used for coupling into and out of the bus waveguide. More details are provided in Section 2 of the SI, including Supplementary Figs. 4, 5.

### CW laser frequency lock

A narrow-linewidth CW laser at telecommunication-wavelength 1556.2 nm (Redfern Integrated Optics, RIO Planex) is split and one part is used for EO comb generation. The other part is frequency-doubled and brought into resonance with the $5S_{1/2}(F_g = 3) - 5D_{5/2}$ $(F_e = 5)$ two-photon transition in $^{85}Rb$ atoms in a microfabricated atomic vapor cell, similar to ref. [88]. A stable frequency lock of the CW laser to the two-photon transition is achieved by means of synchronous detection and by modulating the laser's injection current. Using the same vapor cell and a laser with higher phase noise, we measured the fractional frequency instability of our system to an Allan deviation better than $2 \times 10^{-12}$ for averaging times from 1 to $10^4$ s.

### Miniature atomic frequency source

The 10 MHz sine wave output of a commercial miniature atomic frequency source is provided to the microwave synthesizer as a stable reference. The miniature atomic clock we use is based on a transition in the ground state of $^{85}Rb$. Its relative frequency stability is specified by the manufacturer to be better than $1 \times 10^{-10}$ ($1 \times 10^{-11}$) at 1 s (100 s) averaging time; daily drift $<1 \times 10^{-11}$.

### Mode-scrambler and calibration unit

The mode-scrambler suppresses detrimental laser speckles on the CCD (modal noise) that would result from the coherent LFC illumination after propagating through the multimode fiber train of the SOPHIE spectrograph. It consists of a rotating diffuser disc and has a throughput of ca. 5% effectively washing out any laser speckle pattern. Averaged over one CCD exposure time, it populates all fiber

modes and, therefore, ensures a speckle-free image of the spectrographs's entrance aperture on the CCD. After the mode-scrambler, the light is guided by a multimode fiber to the calibration unit. The calibration unit selects one of the installed calibration light sources or, via a temporarily added additional input port, the light from the LFC. The selected light is then sent to the spectrograph's front-end, mounted at the Cassegrain focus of the OHP 1.93 m telescope, where it is simultaneously injected into A and B fibers feeding the spectrograph[60].

### LFC line fitting and computation of drift

The LFC lines are fitted to find their exact center positions. Due to the limited resolution of the spectrograph, the comb lines are not fully separated but partially overlap, particularly in the UV domain. Therefore, always three LFC lines are fitted together, a central line of interest and two flanking ones to model the flux contributed by their wings. In addition, the diffuse background flux is locally modeled by a polynomial of first order. A detailed characterization of the SOPHIE instrumental line-spread function is not available. Therefore, we have to resort to model all lines with a simple Gaussian shape. In addition, due to the limited sampling (about 2.5 pixels per FWHM) on the detector, a simultaneous and unconstrained fit of all nine model parameters (positions and amplitudes of the three lines in question, one common linewidth, and amplitude and slope for the diffuse background) leads to degeneracies and unphysical fit results. The width of all LFC lines in the fit is, therefore, fixed to an FWHM of 2.47 pixels. This value was inferred from the thorium lines and, although not capturing the variation of the instrumental line-spread function along individual spectral orders or with wavelength in general, was found to better describe the line shapes than a width that is fixed in velocity units.

For each exposure, the fitted line centroids are compared to the corresponding positions in the first exposure of the sequence, the differences converted to radial velocity shifts following the Doppler formula, and then combined to provide one average drift measurement per exposure.

## Data availability

The data shown in the plots have been deposited in the Zenodo database https://zenodo.org/records/12607045; https://doi.org/10.5281/zenodo.12607045.

## Code availability

The numeric simulations in the current study used an open-source code available at https://github.com/pychi-code/pychi.

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

## Acknowledgements

This project has received funding from the Swiss National Science Foundation (Sinergia BLUVES CRSII5_193689 (F.B., D.G., and T.H.)), the European Research Council (ERC) under the EU's Horizon 2020 research and innovation program (ERC StG 853564 (T.H.) and ERC CoG 771410 (V.T.-C.)), and through the Helmholtz Young Investigators Group VH-NG-1404 (T.H.); the work was supported through the Maxwell computational resources operated at DESY. The lithium niobate waveguides were fabricated in the EPFL Center of Micro-NanoTechnology (CMi).

## Author contributions

M.L. built the setup for ultrahigh repetition rate pulse amplification and compression, designed, and tested the waveguides, F.A. developed the waveguide fabrication and poling process and fabricated the waveguides, T.S. developed analysis software and analyzed the calibration data, T.W. operated the microresonator comb and built the EOM setup, T.V. supported laser stabilization, R.B. designed, built, and operated the stabilized CW laser, Z.Y., F.L., and V.T.-C. designed and fabricated the microresonator, M.L., F.A., T.S., T.W., T.V., R.B., V.B., F.B., and T.H. performed the calibration experiments, F.W., F.P., M.A.G., E.O., D.G., S.K., S.L., F.M., B.C., and R.S. supported the experiments, O.H. performed the UV damage study, V.B., L.G.V., F.B., and T.H. conceived and supervised the project. M.L., F.A., T.M.S., and T.H. wrote the manuscript with input from all authors.

## Funding

## Competing interests

The authors declare no competing interests.
