## [Peer Review file · Nature Communications]

REVIEWER COMMENTS

Reviewer #1 (Remarks to the Author):

The authors demonstrate a frequency comb extended down to the near ultraviolet range of 380 nm via cascaded harmonic generation. They use dispersion engineering and period poling to achieve high enough power to perform further experiments. The generation of comb down to the ultraviolet is already a very well studied topic. Nevertheless, it is a very thorough study that shows an impressive application with a device that is challenging to fabricate. The manuscript is well-structured, with state-of-the-art references and presented to be interesting for the broad audience of your journal. Though it suffers from some lack of details to be fully consolidated. I would recommend the following changes and questions to be addressed:

1. The description of the fabrication of the sample is rather short and inaccurate, even in the method section. I would add some supplementary information. Indeed, the periodic poling is not detailed and this is a key aspect to obtain high enough power. With the current description, a reproduction of the experiment is not possible. Please can they provide a more detailed process flow of their fabrication.
2. Figure 1.a is a bit confusing. Is the spectrograph with absorption line from the QSO with metal absorption and H α Ly α forest absorption as they explained from line 60 to line 70? Or it is the mix of light from QSO and exoplanet? Looks like the doppler effect of exoplanet and QSO are different from this figure. Do they want to express that the frequency comb can detect the drift of both QSO and exoplanet? Besides, I can see that they want to show the frequency comb can be used to calibrate the spectrograph. But it looks like the comb is missing the absorption region, is it indicating a redshift?
3. In figure 1.d, what's the different meaning of real line and dashed line?
4. In line 165, the authors explained the band broadening with dispersion. However, since the band broadening is not happened in PPLN waveguides but in ring resonator or other modulation system, why the dispersion still matters here? I would say they may need to have the dispersion as close to zero as possible to obtain a wider QPM range.
5. In figure 2 b, about the higher order QPM, also as explained in line 218, which sum frequency generation is it?
6. In figure 2 defg, could they plot the simulated spectral evolution in wavelength to make it easier to compare with the measured spectra?
7. In figure 2 d, what's the very bright resonance beyond the fourth harmonic at around 830 THz?
8. In the experiments results shown in figure 2, is there also SPM and band broadening for SHG as well since it is so strong?

9. In the experiments results shown in figure 2, how do they estimate the coupling loss to obtain the on chip pulse energy?
10. In the experiments results shown in figure 2, what are the spikes (marked with black box) with input pulse energy of 6.0 pJ?
11. In the experiments results shown in figure 2, is it only fundamental TE mode participated in the band broadening and higher harmonic generation? Is only 1 type of Chi² effect?
12. The fibers used in spectrograph, as mentioned in line 237, are they multi mode in UV to visible range?
13. Are the spectra shown in figure 2 measured by the spectrograph? If not, please indicate the spectrometer used here.
14. In figure 3, if the input of fiber a and b are at different position after the prism as shown in a, why the output of a and b looks exactly the same as shown in b?
15. Can the higher order mode in shorter wavelength from the chip collimated and detected with the spectrograph as shown In figure 3 a?
16. According to figure 4 a, is it should be $\pm 2\text{m/s}$ in 4 hours in line 307?
17. In line 331 to 337, if the measured drift with the reference of thorium emission lines as shown in figure 4 b has the global offset of -10 to -15 m/s, and the scatter in figure 4 b are systematic issue, then which value is used here to conclude the calibration is similar to the test with SOPHIE?
18. The measured drift is 2m/s in 4 hours, which is far from capable to detecting the drift of HI Ly α forest absorption features in QSO spectra with few cm/s over several decades. Along with the test method with systematic error to over 10 m/s. The paper would be more solid if the authors could discuss the ways to improve the stability of LFC and more accurate measurement methods.
19. It would also benefit for the field if the authors could compare the energy consumption and UV conversion efficiency between the three methods shown in the paper (fs laser, fiber comb and ring resonator comb).
20. What's the limit of input power of sub-um PPLN waveguides? For this topic, high harmonic generation to UV comb, is the thin film PPLN really better than diced ridge waveguides (cross sections $> 10\ \mu\text{m}^2$) as they are more robust under high peak power?

Reviewer #2 (Remarks to the Author):

In this paper, important parts of a light source for astronomical spectrograph calibration are demonstrated. The authors show continuum generation at 18 and 25 GHz in nanophotonic lithium niobate waveguides and perform astronomical spectrograph calibration below 400 nm. This is a

critical region of the spectrum that nonlinear fibers have struggled to reliably reach, whether for the production of coherent light for combs or incoherent light for Fabry Perot illumination. The paper is well written, and the overall quality of the paper is good. However, there are still several significant questions that when addressed would make this work stronger and more appropriate for publication in Nature Comm.

Principally, there have been many proof-of-principal demonstrations in this field, and this work falls in that category. While I agree the results are interesting from a nonlinear optics point of view, the impact of this present work on astrophysics and instrumentation appears to be minimal because it is not clear if the current comb improves RV precision or can overcome the challenges associated with actually making RV measurements at or beyond the state-of-the art. The real impact will come after addressing the hard engineering problems, verification of relevant precision limits, and the long-term operation at an actual EPRV spectrograph.

I have a hard time assessing the calibration precision/accuracy from the data of Fig. 4. The data of the drift appear to be inconclusive as to whether the comb is better than the FP—in fact the authors point out that the comb calibration is slightly noisier. And the measurements of the ThAr lines have uncertainty of 10's of m/s that does not push to the state of the art. The fact that ESPRESSO has similar offsets does not build confidence in the calibration, but rather implies a deeper systematic issue independent of the comb. In all of this, the demo precision remains well above the state of the art for EPRV and at least 10-100x above the ultimate need of the EPRV community. That leaves me asking what significant advance is offered by this work if the nonlinear optics is already known and the RV precision is lacking?

One issue is that the repetition rate is not high enough for the spectrograph to fully resolve the comb lines. This makes it unclear if the sub-optimal calibration results arise from the comb or the spectrograph. Only discrete bands of just a few tens of nanometers are available for calibration, leaving a wide swath of spectrum uncovered. Finally, the line intensities are not flattened or stabilized in intensity. The community knows this is a significant issue in frequency comb calibration.

As this was a proof-of-concept test on a spectrograph it might be argued that these issues are technical in nature and can be overcome. But this reader is left wondering about what fundamental advance this paper has enabled, particularly since supercontinuum in thin-film lithium niobate has been previously demonstrated. Beyond that, there remains a significant gap between proof of principle and the requirements of an actual frequency comb that astronomers can use. The results of this paper do not let us know if the engineering issues can be overcome such that the laser comb could be better than a FP, a Th-Ar lamp, or the combination of the two.

Other clarifying points that the authors should address are:

The authors need to strengthen the motivation. Astro spectroscopy at the cm/s level is a very complicated problem and the calibrator is just part of the challenge. Frequency comb calibration is necessary, but not the tall tent pole holding up the discovery of Earth 2.0. Similarly, can the authors provide better evidence that the calibration source is the limitation for the α dot measurements. Same for the Sandage test? What are the known requirements and limitations on the spectroscopy there? Right now, the state of the art in vis spectrographs is at the ~20-30 cm/s level. Will the comb of the authors push this to significantly lower levels?

What is the contribution of the continuum noise from the comb source itself? This is a known problem in astrocombs. From the experimental diagrams and description, it does not appear that there is a filter cavity for noise suppression, something that is common to broadband EO astrocombs generated through chi-3 processes. Chi-2 broadening may have different noise processes than chi-3, but the question remains: is the low contrast truly due to the resolution of the spectrograph or is it indicative of noise washing out the comb lines? In particular, for the higher repetition rate microresonator example, this is not clear in the third harmonic data. The modes of Fig. 5 look well resolved, but there is a significant background flux between the comb teeth (does not go to zero). Can the authors explain this?

Many of the leading spectrographs have combs that are capable of generating UV light. However, the real limitation is their lifetime and not the ability to resolve UV lines (that is simple grating optics). Photodarkening is the primary reason nonlinear fibers are a poor solution at short wavelengths. What can the authors say about this issue? It is unknown if lithium niobate has the same issue, and showing that it does not, especially for the intensities found within nanophotonic waveguides, would be a much stronger result.

The core element of the TFLiN nanophotonic waveguide for UV generation has now been in place for several years. The authors point that out in several places. This includes the benefits and engineering that can be addressed with chirped periodic poling. The authors cite some of the recent papers, but similar designs with chirped poling were introduced in the following works. <https://doi.org/10.1364/NLO.2021.NTh1B.7> and https://doi.org/10.1364/CLEO_QELS.2022.FW4J.2. Also, Ref 47 shows that the chirped poling the authors choose may not be optimal. Can you comment on the choice of poling that increases rather than decreases along the waveguide.

Why is a range of pulse durations (50-150 fs) of the high rep rate pulse train incident on the chip? Factors of 3 in peak power are significant. Can the authors be more specific about what pulse durations are incident in the different situations.

What is the FSR of the FP etalon? Are its modes well resolved? And how does its flux vary across the measured bands in comparison to the comb (which varies by up to ~20 dB). Please report the difference in drift rates measured via the comb and the FP etalon? Are they statistically different?

Please be more specific about how the resolving power of the spectrograph varies as a function of wavelength. And can that be correlated to the modulation depth of the lines as recorded by the spectrograph? In Fig. 3 it appears that modulation depth between 395 and 400 nm is less than that at wavelengths below 390 nm. I would have thought the resolving power of the spectrograph would be lower at the shortest wavelengths, making that the region of the spectrum with the lowest modulation depth. Similar point can be made in the spectra of the 3rd harmonic.

What is the predicted calibration precision based on the photon noise and the number of comb lines employed? And how does that compare to the achieved precision and the design goals of the spectrograph itself. How does that compare for the comb, the FP interferometer and the Th-Ar lamp? That would give information on the projected limitation of this present comb calibration.

Reply to Reviewers

Reviewer report in **black**.

Author reply in **blue**.

Descriptions of revisions are underlined.

We thank both Reviewers for their careful evaluation, the constructive feedback, and their general support of our manuscript. The Reviewers' comments have led to a significantly improved version of the manuscript. Besides major revision to the main text, we added substantial new material in the Supplementary Information (SI) document including a new description of the noise in the $\chi^{(2)}$ -based combs and a new experiment on photo-induced damage in support of our approach. Below we provide a detailed point-by-point reply to all comments raised by the Reviewers and describe the actions that have been taken to address these comments.

Reviewer 1

The authors demonstrate a frequency comb extended down to the near ultraviolet range of 380 nm via cascaded harmonic generation. They use dispersion engineering and period poling to achieve high enough power to perform further experiments. The generation of comb down to the ultraviolet is already a very well studied topic. Nevertheless, it is a very thorough study that shows an impressive application with a device that is challenging to fabricate. The manuscript is well-structured, with state-of-the-art references and presented to be interesting for the broad audience of your journal. Though it suffers from some lack of details to be fully consolidated. I would recommend the following changes and questions to be addressed:

We thank the Reviewer for the careful evaluation and constructive feedback. We were glad to learn that the Reviewer finds our study to be very thorough, demonstrating an impressive application that is of interest to the broad audience of Nature Communications. Further, we agree with the Reviewer that generation of ultraviolet (UV) combs at approximately 100x lower repetition rate is indeed well established. However, for clarity, we would like to stress that the generation of UV combs at > 10 GHz in repetition rate in the UV, and their use in a metrological applications, is unprecedented. In our manuscript, enabled through a periodically-poled nanophotonic waveguide, we show this based on two distinct types of high-repetition rate sources, an electro-optic comb as well as an microresonator combs, both providing naturally high-pulse repetition rate without the problematic spectral filtering of unwanted comb lines. We are grateful for the valuable comments pointing out some lack of detail as well as the insightful questions by the Reviewer. As we will explain in our point-by-point reply below, we have substantially revised our manuscript to address these comments and questions.

1. The description of the fabrication of the sample is rather short and inaccurate, even in the method section. I would add some supplementary information. Indeed, the periodic poling is not detailed and this is a key aspect to obtain high enough power. With the current description, a

reproduction of the experiment is not possible. Please can they provide a more detailed process flow of their fabrication.

We appreciate this request by the Reviewer as indeed the fabrication of the waveguides relies on novel techniques that are not yet established in the field. To address this point we have created the new Section 2 in the Supplementary Information (SI), describing and illustrating in detail the process flow of the fabrication (including periodic poling).

2. Figure 1.a is a bit confusing. Is the spectrograph with absorption line from the QSO with metal absorption and HI Ly α forest absorption as they explained from line 60 to line 70? Or it is the mix of light from QSO and exoplanet? Looks like the doppler effect of exoplanet and QSO are different from this figure. Do they want to express that the frequency comb can detect the drift of both QSO and exoplanet? Besides, I can see that they want to show the frequency comb can be used to calibrate the spectrograph. But it looks like the comb is missing the absorption region, is it indicating a redshift?

We thank the Reviewer for pointing out confusing elements in Figure 1a, likely resulting from its very condensed design. QSO and exoplanets represent two distinct and separate science cases. As the Reviewer correctly assumes, they can both be addressed through astronomical precision spectroscopy, hence benefiting from a comb calibrated spectrograph. However, the two science cases are not pursued simultaneously (and QSO light and light from an exoplanet are never mixed). In the cosmological science cases one aims at measuring the cosmological red-shift of e.g. Ly α absorption lines. In the case of exo-planets, one aims at measuring stellar spectra (specifically, absorption features in a stellar atmospheres), which can experience Doppler-frequency shift (periodically alternating red- and blue-shift) if a planet is orbiting the star. To measure the cosmological red-shifts or the Doppler-frequency shifts caused by exoplanets one needs in both cases a precise calibrator. The spectrally-resolved comb lines can provide this calibration. The comb lines, although they never interact with the astronomical object (hence no absorption), can be sent to the astronomical spectrograph and used for frequency calibration. Often astronomical light and frequency comb light are sent simultaneously into the spectrograph through two different entrance aperture so they can be recorded and compared directly on the spectrograph. To clarify this, we took several measures: (1) Figure 1a has been revised to show the two science cases separately (i.e., two telescopes, no mixing of light), the labeling in the illustration of the spectrograph has been improved to indicate that the absorption feature is only contained in the astronomical spectrum, whereas the comb and its lines, are a local calibration source without any absorption features. (2) In the main text in the context of extending astrocombs towards shorter wavelength, we now explicitly mention the exo-planet science as a separate case from the cosmological science cases.

3. In figure 1.d, what's the different meaning of real line and dashed line?

We thank the Reviewer for pointing out this information missing from the figure caption. First, for clarity, the horizontal continuous lines labeled $\chi^{(3)}$ indicate spectral broadening through third-order nonlinear effects (mostly four-wave mixing/self-phase modulation). However, the Reviewer is likely referring to the curved continuous and dashed lines, which were intended to indicate different $\chi^{(2)}$ -nonlinear frequency conversion processes: Continuous-lines indicated 'harmonic generation' processes, namely of the fundamental comb, as well as, the second harmonic of the second harmonic comb. Dashed lines indicated 'sum frequency generation' processes, namely those of the fundamental comb with the second harmonic comb, and of the fundamental comb with the second harmonic comb. As strictly speaking also the harmonic generation processes rely on sum-frequency generation, we

do realize that this representation may have been confusing. To avoid confusion, we have simplified Figure 1d and now only indicate which parts of the spectrum are dominated through $\chi^{(2)}$ - and $\chi^{(3)}$ -nonlinear processes, respectively. Moreover, we have now labeled the different spectral regimes as “fundamental”, “second-harmonic”, “third-harmonic”, “fourth-harmonic”, which also makes a better connection to the remaining figures.

4. In line 165, the authors explained the band broadening with dispersion. However, since the band broadening is not happened in PPLN waveguides but in ring resonator or other modulation system, why the dispersion still matters here? I would say they may need to have the dispersion as close to zero as possible to obtain a wider QPM range.

We thank the Reviewer for this important question. Indeed, the Reviewer is correct that if we were only interested in $\chi^{(2)}$ -nonlinear processes, a sensible design approach would be to target a dispersion as close to zero as possible for both fundamental and harmonic wavelength as to maximize the QPM bandwidth for one specific poling period. Although this would be a sensible design approach, this is far from possible over the frequency interval we are considering here due to material intrinsic dispersion (wavelength dependence of the refractive index) and multiple poling period (e.g. a chirped poling pattern) is critically required to achieve QPM over a wide wavelength range. In turn, the ability to implement such poling patterns our fabrication process permits leveraging the waveguide’s dimension and its group velocity dispersion (GVD) as a free-design parameter (as the $\chi^{(2)}$ processes do no longer require a specific waveguide dispersion). Indeed, we use this to design the waveguide’s dispersion such that they also support additional $\chi^{(3)}$ -based nonlinear broadening inside the waveguide, further extending the spectra of the high-repetition rate input sources. The effect of this additional broadening is visible, for instance, in the simulations shown in Figure 2d (bottom), where the fundamental spectrum (and thus also the harmonic spectra) each broaden significantly beyond their initial width. Curiously, the broadening effect is further enhanced in the presence of a poling and quasi-phase matching as can be seen by comparing Figure 2d and Figure 2e. This can be explained through the efficient $\chi^{(2)}$ -nonlinear interactions, which, in a two-step process, can boost the effective $\chi^{(3)}$ -nonlinearity (For instance a SHG photon can generate signal and idler photons around the initial pump, akin to those generated in a direct $\chi^{(3)}$ -mediated four-wave mixing process). To improve the manuscript in this point we now explicitly describe in the main text (1) the need for poling regardless of waveguide geometry, (2) that an anomalous GVD waveguide can provide additional spectral broadening to the input spectra, and (3) mention that cascaded phase-matched $\chi^{(2)}$ processes can significantly contribute to a larger effective $\chi^{(3)}$ -nonlinearity, supporting efficient broadening.

5. In figure 2 b, about the higher order QPM, also as explained in line 218, which sum frequency generation is it?

We appreciate the Reviewer’s request for clarification regarding higher-order QPM. Indeed, higher-order QPM is a collective term for approaches where the spatial modulation (e.g. poling) of the $\chi^{(2)}$ -nonlinearity occurs not with the ideal period Λ (cf. Equation 2) but with a multiple of this period. For instance, one can achieve the same nonlinear effect by utilizing an odd integer q multiple of the poling period (in our case $q = 3$). In this way the physical dimension of the poling structure becomes larger and the fabrication requirements are hence relaxed. For completeness, we note that under the condition that the poling is not symmetric (i.e. the two domains in one poling period are not of equal length), also even q contribute to higher-order QPM; due to fabrication imperfection, this effect will likely also be present to some extent. To better describe high-order QPM we have added a clarification along the lines above to the manuscript. Moreover, we have added the new reference

Suhara et al., (“Theoretical Analysis of Waveguide Second-Harmonic Generation Phase Matched with Uniform and Chirped Gratings”, IEEE J. of Quantum Electronics, 26, 7, 1990) that provides a detailed analysis of higher-order QPM in the analogous context of second harmonic generation.

With regard to Figure 2b, we concur with the Reviewer that the original representation can be improved. The original Figure 2b showed the ideal QPM periods Λ for SHG, SHG + Fundamental, THG + Fundamental and SHG+SHG and, in addition, the curves for select higher order processes. To improve clarity, we now added all higher-order (for $q = 3$) QPM curves (Λ multiplied by 3) to Figure 2b, i.e., each higher-order QPM curve corresponds to one with 3-times lower period, phase matching the same process.

6. In figure 2 defg, could they plot the simulated spectral evolution in wavelength to make it easier to compare with the measured spectra?

We agree with the Reviewer that the wavelength axis may be more familiar to many. However, in our manuscript we consciously decided to use the frequency axis as the main x-axis at the bottom, with wavelengths indicated in addition at the top. The reasoning for choosing the frequency axis as the main axis is that for the very broad spectra we present and in particular spectra with multiple harmonics, only the frequency axis provides a proper representation of the span and the intrinsic nature of harmonics, in the sense that the harmonic appear equidistantly spaced in frequency but not in wavelength. Higher harmonics plotted in wavelength start to be squeezed together, resulting also in bad readability. To allow for easy comparison of experimental and simulation data in Figure 2defg, we have added to the figure caption the previously missing information that the plots of experimental data and simulation data in Fig.2 defg share the same x-axes, indicating frequency at the bottom and wavelength at the top. We thank the Reviewer for identifying this missing information.

7. In figure 2 d, what’s the very bright resonance beyond the fourth harmonic at around 830 THz?

We appreciate the interesting observation by the Reviewer of a pronounced peak in the fourth harmonic in figure 2d. This feature can be attributed to a higher-order quasi-phase matching condition within the waveguide: Our analysis reveals an overlap between the quasi-phase matching conditions for second harmonic generation (SHG) and the generation of the fourth harmonic (FHG). Specifically, both the higher order SHG+SHG and THG+fundamental quasi-phase matching conditions overlap near a fundamental wavelength of roughly 1440 nm, as evidenced by the intersection of the corresponding quasi-phase matching curves with the SHG curve around this wavelength in figure 2b. The required poling period is around 4 μm . The pronounced slope of the FHG quasi-phase matching curves explains the wavelength selectivity and the relatively enhanced conversion efficiency across a relatively narrow bandwidth. This specificity is also revealed by the presence of distinct dips in the spectral intensity of the second and third harmonics around 720 nm and 480 nm that coincide with the emergence of the strong signal observed by the Reviewer. These features highlight the efficient energy transfer from these harmonic orders directly into the fourth harmonic.

8. In the experiments results shown in figure 2, is there also SPM and band broadening for SHG as well since it is so strong?

We thank the Reviewer for this question. While in principle the SHG could be broadened through its own self-phase modulation (SPM) this contribution is here not the dominating effect. Instead through the choice of anomalous GVD at the pump wavelength (as described above), the fundamental input spectrum (a short input pulse) experiences spectral broadening, mostly through SPM. This

broadening then transfers to the SHG, THG and FHG spectra that are driven by the fundamental spectrum. The absence of a dominating direct SHG ‘self-broadening’ (through SPM) is plausible considering that the SHG signal, although efficiently generated, is still noticeably weaker than the fundamental pump signal. Moreover, while the efficient SHG generation is enabled by QPM, this does not lead to a match of the group velocities and the SHG signal will thus likely not be a short high intensity pulse (in contrast to the pump signal), which could efficiently drive SPM. Additionally, the group velocity dispersion (GVD) is strongly normal at the SHG frequency, limiting SPM broadening of the SHG by the SHG itself. To clarify this question, we now explain in the main text that the broadening occurs at the fundamental wavelength and is then transferred to shorter wavelength via $\chi^{(2)}$ -effects.

9. In the experiments results shown in figure 2, how do they estimate the coupling loss to obtain the on chip pulse energy?

In detail, to perform the input coupling, we use a high-numerical aperture (NA=0.7) aspheric glass lens for a spot size compatible with the mode size at the input facet. The output coupling in turn relies on an off-axis parabolic mirror with an NA of roughly 0.4 for collimation. The spectra in Figure 2 were captured with a fluoride multi-mode fiber attached to a reflective collimator that collects the collimated chip output. We particularly chose free-space optics to facilitate power sweeps of the driving pulses while keeping the pulse duration unaffected.

In our chip coupling configuration the exact quantification of coupling efficiencies is not trivial due to the use of an asymmetric coupling setup and free-space optics: Initially we used a conservative (too high) estimate of 17.5% for the input coupling efficiency, preventing us from claiming too low values for the required pulse energy.

To address the Reviewer’s question, we have performed a more accurate characterization: First we measure the coupling efficiency for a lensed fiber in a symmetric lensed fiber configuration. We then use a lensed fiber at the chip’s output to determine the input coupling efficiency of the high-NA lens. We find $13 \pm 2\%$. We updated the values in Figure 2 accordingly and added a description of the coupling setup and the determination of the input coupling efficiency to the main text.

10. In the experiments results shown in figure 2, what are the spikes (marked with black box) with input pulse energy of 6.0 pJ?

We are not certain, if we correctly understand this question, as we are not sure what the Reviewer means by “marked with black box”. First, we note for clarity that, within each experimental data panel, the geometric shape (black box?) indicate the trapezoidal waveguide cross-section. To make this point clear, we now explicitly clarify this in the figure caption. Now, concerning the very good observation and what we understand to be the question of the Reviewer regarding the spikes for pulse energy of 6 pJ (yellow trace): For very low pulse energy the signal of the THG and FHG are extremely weak and the noise floor of the utilized optical spectrum analyzer (OSA) becomes visible and manifest itself as a spurious spiky signature on the logarithmic y-axis. This also explains why the spikes are present outside the core spectral interval of the harmonics. Once the pulse energy is increased all harmonics significantly exceed the noise floor and the spurious signals are no longer present. To clarify the origin of the spikes, we have added an explanation to the caption of Figure 2.

11. In the experiments results shown in figure 2, is it only fundamental TE mode participated in the band broadening and higher harmonic generation? Is only 1 type of Chi2 effect?

We thank the Reviewer for bringing up this non-trivial point. Indeed, we design our structures for a type 1 interaction of purely transverse electric (TE) polarization. These processes are expected to give broadband harmonic spectra as we confirm experimentally. However, we can not exclude that type 2 interactions involving also the orthogonal polarization provide additional contributions to the spectra. Similarly, while we design the waveguides to already work reliably with only the fundamental transverse waveguide mode, higher order transverse waveguide mode may also contribute to the overall spectral generation. To clarify this point we now mention the possible contribution of the orthogonal polarization and higher-order transverse modes.

12. The fibers used in spectrograph, as mentioned in line 237, are they multi mode in UV to visible range?

We thank the Reviewer for requesting clarification of this important technical aspect. Indeed, to transport the generated UV and visible light from the waveguide to the spectrograph, we utilize only fibers that are multi-mode across the entire relevant UV, visible and infrared regimes (core diameters ranging from 100 μm to 400 μm). Specifically, to connect the comb source to the spectrograph input interface, we utilize a fiber with a core diameter of 220 μm . A description of the fiber routing after the spectrograph interface is given by Perruchot et al. (“The SOPHIE spectrograph: design and technical key-points for high throughput and high stability”, Ground-based and Airborne Instrumentation for Astronomy II; 70140J, 2008). We modified the Methods section in the manuscript to clearly state that all fibers used after the waveguide are multi-mode, and added the mentioned reference in this context. In addition, we now also indicate in Figure 3a that the fibers following the lithium niobate chip are multi-mode.

13. Are the spectra shown in figure 2 measured by the spectrograph? If not, please indicate the spectrometer used here.

We thank the Reviewer for pointing out the missing information regarding the spectrometer used for the measurement in Figure 2. Indeed, those spectra were *not* measured by the astronomical spectrograph but obtained in preparatory experiments during the waveguide design and initial testing phase. The spectra in Figure 2 have been measured with a grating based optical spectrum analyzer (OSA, model Yokogawa AQ6374) and, as mentioned above, the light has been coupled to the spectrum analyzer via reflective optics (off-axis parabolic and regular mirrors) / via a fluoride multimode fiber. We have added this previously missing information to the main text and the caption of Figure 2.

14. In figure 3, if the input of fiber a and b are at different position after the prism as shown in a, why the output of a and b looks exactly the same as shown in b?

We appreciate the question and realize that our original manuscript did not sufficiently clarify the role of the fibers A and B. The SOPHIE spectrograph is a cross-dispersing spectrograph. Light is coupled to the spectrograph through an entrance slit and then cross-dispersed onto a 2D-dimensional detector array. This is accomplished through the combination of two ‘crossed’ dispersing elements (here: a prism and an Echelle-grating), arranging the spectrum on the detector in a vertically separated set of nearly horizontal lines (*Echelle orders*), where each line represents a distinct wavelength interval. The arrangement of the wavelength on the spectrograph is conceptually shown in the small inset in Figure 3b. As many high-resolution spectrographs, the SOPHIE instrument is fed by 2 optical fibers (A and B fiber). While in principle one fiber would be enough to perform spectroscopy, the dual-fiber configuration enables simultaneous recording of an astronomical target spectrum and the

spectrum of a calibration light source. The fibers' output facets are placed next to each other on the spectrograph entrance slit, each creating a cross-dispersed spectrum on the 2-dimensional detector. On the detector, the two spectra are offset (here: vertically) corresponding to the offset of the fibers on the spectrograph slit. This offset is smaller than the offset between consecutive Echelle-orders so that no overlap between different Echelle-orders of A and B fibers exists. For our experiment, the frequency comb source is split evenly in power and injected in both A and B fibers, resulting in two strands of slightly vertically offset resolved calibration lines (e.g. magnified insets in Figure 3b). To clarify the role of A and B fiber and to better explain how they are manifest in the detector data in Figure 3b, we have added a description of the working principle of a cross-dispersing spectrograph to the main text. Moreover, we now explain in the main text the splitting and injection of the comb source into both fibers and the resulting pattern on the detector. In addition, we have modified Figure 3a to better represent the placement of Fiber A and B within the setup.

15. Can the higher order mode in shorter wavelength from the chip collimated and detected with the spectrograph as shown In figure 3 a?

Indeed, as the Reviewer alludes to, coupling the light from the waveguide into the spectrograph is non-trivial. In principle, we could directly couple the entire output of the waveguide, including higher-order spatial modes at short wavelength, to a multi-mode fiber via butt-coupling with good efficiency. However, the detector would then receive significant fundamental pump and second harmonic light that could lead to unwanted saturation, measurement artifacts, or even damage of the detector. To suppress the fundamental and SHG light, we approximately collimate and slightly focus the light from the waveguide with an off-axis broadband parabolic mirror. Next the light passes through a prism for coarse wavelength selection. This permits us to suppress SHG and fundamental light, as well as, to equalize the intensities of third and fourth harmonic by intentional misalignment. Finally, the selected wavelength interval including light of third and fourth harmonic enters the core of a multi-mode fiber. Using a multi-mode fiber with large core diameter (here 100 μm) is key, as it enables the efficient collection of the incoming light including contributions from higher order modes. We estimate the waveguide to multi-mode fiber coupling efficiency to be on the few percent level, which could likely be improved. However, as the achieved transmission of the calibration light to the spectrograph was found to be more than sufficient, we made at this point no attempt of improving it further. To clarify this point, we have added a more detailed description in the main text and we also modified Figure 3a to more accurately represent the setup and, as described in the previous point, to indicate the multi-mode fiber.

16. According to figure 4 a, is it should be $\pm 2\text{m/s}$ in 4 hours in line 307?

We thank the Reviewer for carefully cross-checking data and description. We agree that the short-term fluctuations of the data shown in Figure 4a are approximately positive and negative (the unit m/s corresponds to the equivalent Doppler-shift velocity). What we meant by stating "ca. 2 m/s over 4 hours" is the *systematic* drift of the signal from 0 to -2 m/s that occurs over the measurement interval of 4 hours. We have modified the corresponding statement in the manuscript to highlight the systematic nature and the specific (here negative) sign of the systematic drift.

While all presented measurements in Figure 4a show certain short-term scatter dominated by measurement imprecision, they all detect consistently the underlying systematic drift of the spectrograph. The consistent detection of the systematic drift is important as the agreement (within the measurement precision) between the third harmonic comb and the Fabry-Perot cavity validates the calibration via the third harmonic comb and agreement between third and fourth harmonic validates the calibration with the UV portion of the comb (where no Fabry-Perot cavity calibration is avail-

able). To improve the clarity regarding this point, we have improved the corresponding section in the manuscript.

17. In line 331 to 337, if the measured drift with the reference of thorium emission lines as shown in figure 4 b has the global offset of -10 to -15 m/s, and the scatter in figure 4 b are systematic issue, then which value is used here to conclude the calibration is similar to the test with SOPHIE?

We thank the Reviewer for this question, which we believe may have emerged from a misunderstanding of what is shown in Figure 4b. In contrast to the 4 hour long measurement of the spectrograph drift in Figure 4a (i.e. by how much does the frequency calibration change over time), Figure 4b investigates how accurate the frequency comb based calibration is at a given moment in time. To this end, we perform two short measurements with the spectrograph of (i) a Thorium-Argon (ThAr) lamp, and (ii) of the frequency comb. Both measurements are taken in quick succession, so that there is virtually no drift of the spectrograph. We then derive from the discrete comb lines with the frequencies given by Equation 1 the so called *wavelength solution*. This wavelength solution can be seen as an interpolation between the comb lines and permits assigning a frequency (or wavelength) to every position on the detector. Based on this we can then assign frequencies to each detected ThAr line. To validate the accuracy of the comb-based calibration we compare the so obtained ThAr line frequencies to the cataloged ThAr line frequencies on a line-by-line basis. Figure 4b shows the deviation of this line-by-line comparison in units of m/s (equivalent Doppler-shift velocity) plotted over the cataloged wavelength. As the projection histogram (Figure 4b, right panel) shows, the obtained deviations are consistent with a zero-offset, implying that the comb-based calibration is accurate to the level testable with the spectrograph. To improve clarity, we have expanded the description of the measurement presented in Figure 4a and Figure 4b in the main text. We also improved the explanation of the wavelength solution in the main text and in the Methods.

18. The measured drift is 2m/s in 4 hours, which is far from capable to detecting the drift of HI Ly α forest absorption features in QSO spectra with few cm/s over several decades. Along with the test method with systematic error to over 10 m/s. The paper would be more solid if the authors could discuss the ways to improve the stability of LFC and more accurate measurement methods.

We appreciate this important comment by the Reviewer and agree that this discussion was missing in the previous version of the manuscript. Before we provide a discussion on how the LFC can be improved we would first like to clarify two points with regard to the data shown in Figure 4 to avoid confusion:

- *Concerning the systematic drift of 2 m/s in 4 hours as observed in Figure 4a:* Stabilizing a spectrograph, in particular on long timescales, to the level required for detecting e.g. the RV signal caused by an exoplanet or the extremely small cosmological redshift drift is virtually impossible. However, this is also not necessary. The solution to this problem, pioneered e.g. by Mayor et al. at OHP, is to accurately track the drift of the spectrograph using a simultaneous wavelength calibration. For this, light from an astronomical source under investigation (star or QSO) is fed to Fiber A of the spectrograph, while light from a highly stable calibration light source is simultaneously injected into Fiber B. This allows to measure the spectrograph drift and to correct for it. The question is therefore not by how much a spectrograph drifts, but how accurately one can track it. This clearly requires suitable, highly stable calibration light sources. Historically, Thorium-Argon (ThAr) hollow-cathode lamps have been used for this, but at some point were often replaced by passively-stabilized white-light Fabry-Pérot etalons, which itself have to be calibrated against e.g. ThAr lamps. While showing good performance, their stability, in particular on multi-year timescales, is limited. Pushing the limits and observing smaller RV

shifts in astronomical objects over longer timescales therefore requires LFCs, which are locked to a fundamental time standard and therefore free of any long-term drift. Regarding the SOPHIE spectrograph in our case, it was not initially designed for such precision measurements and the observed drifts are expected. We demonstrate in Fig. 4 that we can indeed track the drift of the spectrograph with good accuracy and that after LFC-based drift correction the RV measurements of a celestial source would be substantially more stable. State-of-the-art spectrographs like HARPS or ESPRESSO, and upcoming UV-capable instruments (ANDES) are indeed significantly more stable, however, even these can exhibit m/s drifts over several days and also for those fore-front instruments, simultaneous calibration by a highly-stable wavelength calibration source is vital and essential.

- *Concerning the systematic error of ≈ 10 m/s obtained in the comparison between the established ThAr standard and the LFC in Figure 4b:* Although we find mean values in our histograms that are not zero, the observed values are less than half a standard deviation away from zero and statistically compatible with a null result. It is nevertheless interesting to note the deviation from zero is not observed for the first time and an offset from -10 to -15 meters has been reported based on a similar comparison between ThAr and a LFC of very different design on the ESPRESSO spectrograph (Schmidt et al., *Astronomy & Astrophysics* 646, A144 (2021)); although interesting, an investigation of this matter is not relevant to the present work. Finally, we note for clarity, that high accuracy and precision can only be reached statistically when a large number of spectral features and calibration markers are available. Although the frequency comb provides thousands of comb lines (cf. Fig. 3), the ThAr lamp only emits a rather sparse spectrum of ca. 100 lines in the relevant frequency range. It is this low number of ThAr lines that prevents us to obtain a smaller standard deviation in our comparison in Figure 4b. This does however not imply that the accuracy of the comb based calibration is limited to that level.

We would like to emphasize that we do *not* claim that the current system can reach a calibration on the cm/s level. In Figure 4a, we demonstrate that the drift measurement obtained from the two parts of the LFC are consistent with each other and also consistent with the drift measurement obtained from the FP. This validates the LFC as suitable calibration source. The remaining scatter in the drift measurement is on the scale of 0.5 to 1 m/s. This approximately marks the precision achievable here. Reaching the needed precision is a community effort involving advances in spectrographs, data analysis and calibration sources. Some of these aspects are outlined in the ELT/ANDES Whitepaper (Martins et al. 2023, doi:10.48550/arXiv.2311.16274, new reference in the manuscript). Our work demonstrates for the first time spectrograph calibration in the UV with a frequency comb, which represents a key milestone in these developments. Importantly, the UV comb is derived from robust telecom wavelength high-repetition rate sources (electro-optic comb and microresonator), and does not require mode-filtering of lower repetition rate comb (which is known to be problematic due parametric regrowth of suppressed comb lines).

Now, finally directly addressing the Reviewer’s comment, there is a clear route towards further improvements in future work:

- One practical limitation regarding higher precision in our demonstration resulted from the modest resolving power of the SOPHIE spectrograph compared to our comb’s line spacing: While the comb lines are well separated, their tails still impact neighboring lines, which complicates the data analysis and gives rise to elevated uncertainties. Finding the sweet-spot between high line density and line separation is non-trivial and was not the goal in this proof-of-concept demonstration of UV calibration. More recent and upcoming spectrographs achieve significantly higher resolution so that this limitation is not expected to persist. Should nevertheless a higher repetition rate be desired, we emphasize that implementing a higher repetition rate

comb source is readily feasible and compatible with our approach. In our case, operating with higher repetition rate was not possible due to the specific optical components available to us at this point (maximal electro-optic modulation frequency 18 GHz and re-microresonators with fixed 25 GHz line spacing). We have revised the manuscript to clearly point out the fitting of the overlapping comb lines as limitation that can be overcome with higher repetition rate or spectrographs with higher resolving power.

- Another improvement, which was not the goal of our proof-of-concept demonstration, is extending the bandwidth of each harmonic and potentially closing the gaps between the harmonics, as it was demonstrated in Figure 2d,g for low repetition rate. The then larger number of comb lines would lead to a statistical improvement of the calibration precision (scaling with \sqrt{N} , where N is the number of comb lines). Here, the main challenge is avoiding overlapping harmonics with different offset frequencies (cf. Eq. 1), which would result in an inconsistent frequency comb. For a consistent comb the comb’s central frequency must be an integer multiple of the repetition rate $\nu_0 = n f_{\text{rep}}$ (i.e. the carrier-envelope-offset frequency should be zero). Practically, this can be achieved for instance by referencing the central driving laser to an auxiliary low-repetition rate frequency combs similar to Obrzud et al. (“Broadband near-infrared astronomical spectrometer calibration and on-sky validation with an electro-optic laser frequency comb”, Optics Express 26, 26, 34830, 2018), or by zeroing the beating signal between overlapping harmonics as described by Okubo et al. (“Offset-free optical frequency comb self-referencing with an f-2f interferometer”, Optica 5, 2, 288 (2018)). We have revised the manuscript to not only point out this challenge but to also describe the possible solutions; both mentioned references appear in this context. Another aspect when going towards wider span is the suppression of repetition rate phase noise in electro-optic combs, which can, for very broadband spectra, result in a complete loss of contrast for the comb lines. In our demonstration the contrast remains sufficiently high, but significant spectral extension will necessitate phase noise suppression. The established approach, pioneered by Beha et al. (“Electronic synthesis of light”, Optica 4, 4, 406, 2017, a reference in our manuscript), to achieve phase-noise suppression is sending the electro-optic modulation comb through a Fabry-Perot filter cavity. We have added discussion regarding phase noise to the main text and also added an extensive new section in the SI (Section 1) that presents a detailed analysis of the phase-noise and its impact on comb lineshape, which for $\chi^{(2)}$ -based combs hasn’t been studied before to our knowledge. This analysis confirms the validity of our current approach and also the prospect for more broadband spectra. Finally, to show that more broadband spectra are indeed feasible, we present in the new Section 5 of the SI a broadband spectrum that closes the gaps between the harmonics and that we generated with our electro-optic comb generator when optimizing the pulse duration to reach below 50 fs. However, before this spectrum can be employed for calibration the improvements presented above will need to be implemented.
- Regarding the nanophotonic lithium niobate chips, a major step forward can be expected through the improvement of coupling efficiency. In our case, the chips used simple tapered input couplers with a width at the facet of $2.5 \mu\text{m}$ (This information was added to the manuscript), resulting in a characteristic input coupling efficiency, depending on coupling method, of ca. 20%. From other platforms, such as silicon nitride waveguides, we know that advanced coupler designs (e.g. via narrow inverse tapers, 3D etching or second waveguide layers) can enable coupling efficiency that are at least three times higher, resulting in a corresponding reduction of the power requirement by the same factor. Reduced optical power requirements would simplify the system’s architecture (e.g. amplification of short pulses to high average power), and provide easier access to more broadband spectra. This is discussed within the revised conclusion.
- Finally, spectral flattening of the comb’s spectral envelope would avoid under or over saturated

areas on the spectrograph for optimal line extraction and reduction of systematic effects that can result from strong intensity gradients in the comb’s envelope. Such spectral flattening can be implemented via spatial light modulators, which can also operate in the UV wavelength range. As our comb spectrum provides more than enough power, spectral flattening should be readily possible. We now mention spectral equalization as one of the next steps towards improved systems in the conclusion of the main text.

In summary, the goal of our work is opening a pathway to spectrograph calibration in the UV, which previously has not been achieved. We demonstrate how the challenging UV region can be accessed directly with mature telecom-wavelength high-repetition rate laser technology via nanophotonic waveguides. This is a breakthrough result, however, we concur with the Reviewer that additional improvements will be needed before the novel approach can, in combination with advancement in telescopes, spectrographs, astrophysics and data analysis, enable cm/s-level precision measurements on astronomical targets. We hope that the extensive discussion above and the revision in the manuscript, address this important comment by the Reviewer and convincingly convey the potential of the new approach presented in this manuscript.

19. It would also benefit for the field if the authors could compare the energy consumption and UV conversion efficiency between the three methods shown in the paper (fs laser, fiber comb and ring resonator comb).

We appreciate this comment regarding energy consumption and conversion efficiency, which we had not discussed in the initial version of the manuscript.

- *Concerning energy consumption:* The energy consumption of the electro-optic comb generator is dominated by the multi-W erbium-doped fiber amplifier (EDFA) and the microwave electronics. Assuming a 10% wall-plug efficiency the EDFA will approximately consume 35 W of electrical power. The microwave synthesizer consumes under maximal load, depending on the specific hardware, ca. 80-130 W of electrical power, the microwave amplifiers jointly ca. 50 W of electrical power. In sum this amounts to approximately 200 W of electrical power consumption, less than a standard desktop computer. In case of the microresonator-based approach, one can subtract the power consumption of the microwave source and microwave amplifier, resulting in a power consumption of approximately 35 W. The power consumption of our mode-locked laser is below 15 W. However, considering the ~ 200 -times lower repetition rate, the energy consumption per pulse is not better than of the electro-optic comb, although it reaches higher per pulse energy. To address this point, we now provide this discussion on power consumption in the SI, Section 3.1.
- *Concerning conversion efficiency:* The conversion efficiency is defined by the input pulse parameters and independent of the repetition rate (and in that independent from the source). The reason for that is that the spacing between consecutive pulses is larger than their temporal extent (even after dispersion and nonlinear effects). Based on the spectra experimentally recorded in Figure 2, we can estimate the conversion efficiency as a function of the pulse energy for a fixed pulse duration of 80 fs. We find that for input pulse energy above 60 pJ the conversion efficiency into any harmonic well exceeds 0.1%, which is far more than what is needed for spectrograph calibration. To address this point, we have added a detailed description on the conversion efficiency and its measurement to the SI, Section 3.2.

20. What’s the limit of input power of sub- μm PPLN waveguides? For this topic, high harmonic generation to UV comb, is the thin film PPLN really better than diced ridge waveguides (cross sections $> 10 \mu\text{m}^2$) as they are more robust under high peak power?

We thank the Reviewer for these questions regarding limits in input power and differences between nanophotonic waveguides and conventional large mode-area ridge-waveguides.

- *Concerning the limit input power:* In our case, we operated the nanophotonic waveguides with input power levels of up to 3 W of average power (before the chip) without noticeable damage. We expect that higher power levels are feasible. However, as higher power levels were not needed in our case, we did not investigate this systematically. With improved coupling efficiency, as alluded to in a reply to a previous point, the needed power levels may even be reduced further. To clarify this point, we now state in the main text that we did not observe damage from too high optical input power in the experiments presented in this manuscript.
- *Concerning the comparison between nanophotonic waveguides and conventional large mode-area ridge-waveguides:* We believe that the two geometries are complementary, each offering different advantages. In terms of absolute power handling and absolute output achievable output power large mode-area waveguides are superior, simply for the reason that they can sustain (but also require) higher photon fluxes. For instance in the field of high-power THz wave generation, lithium niobate waveguides with large cross-section are used. In our case, high power levels were not the goal but rather the ability to engineer the GVD to facilitate additional spectral broadening on the lithium niobate chip, as discussed above. In this case small-cross section waveguides are advantageous as they provide this degree of freedom through geometric waveguide dispersion. This would not be possible in a large mode-area waveguide. To address this point we now mention in the main text that GVD engineering is possible due to the tightly confining core and the geometric dispersion of our waveguides.

In conclusion, we again thank the Reviewer for the careful review and the valuable comments, which have led to a substantially expanded and improved manuscript. We hope that in light of our revision and explanations, the Reviewer can support publication of this manuscript.

Reviewer 2

In this paper, important parts of a light source for astronomical spectrograph calibration are demonstrated. The authors show continuum generation at 18 and 25 GHz in nanophotonic lithium niobate waveguides and perform astronomical spectrograph calibration below 400 nm. This is a critical region of the spectrum that nonlinear fibers have struggled to reliably reach, whether for the production of coherent light for combs or incoherent light for Fabry Perot illumination. The paper is well written, and the overall quality of the paper is good. However, there are still several significant questions that when addressed would make this work stronger and more appropriate for publication in Nature Comm.

We thank the Reviewer for the careful evaluation and constructive feedback. We were glad to learn that the Reviewer recognizes the significance of our work demonstrating an important part of a light source for astronomical spectrograph calibration in a critical region of the spectrum that previous fiber-based approaches have struggled to reliably reach. We also acknowledge that the Reviewer identifies several significant questions that need to be addressed. In our point-by-point reply below, we address all comments and detail the revision and additional experiments that have been carried out in response to the Reviewer. To organize our reply, we have added numbers (1), (2), .. to each point.

(1) Principally, there have been many proof-of-principal demonstrations in this field, and this work falls in that category. While I agree the results are interesting from a nonlinear optics point of view, the impact of this present work on astrophysics and instrumentation appears to be minimal because it is not clear if the current comb improves RV precision or can overcome the challenges associated with actually making RV measurements at or beyond the state-of-the art. The real impact will come after addressing the hard engineering problems, verification of relevant precision limits, and the long-term operation at an actual EPRV spectrograph.

We thank the Reviewer for this comment and agree that the core advancement presented in our manuscript is in the field of nonlinear optics and frequency comb technology. Through a novel approach, we demonstrate for the first time calibration of an astronomical spectrograph in the UV with a high-repetition rate astrocomb. Importantly, this astrocomb derives from mature telecom wavelength sources and does not require suppression of unwanted comb lines (such suppressed lines are known to be problematic if they re-grow through nonlinear effects). This is unprecedented in this wavelength range.

Regarding the potential of the current source to improve radial-velocity (RV) measurements, several aspects can be considered:

First, we note that one essential component to high RV precision is a large spectral interval that is covered by the spectrograph and the calibrator; high precision is obtained statistically through a high number of spectral features. Our work demonstrates a significant extension of the previously accessible spectral interval into a *new* wavelength regime, hence it creates opportunities to improve RV precision. While the span of the frequency interval covered in our work does by itself not exceed the span of previous demonstration, our system can, already in its current form, serve as a wavelength range extender, in conjunction with a conventional system, leading to an improved RV precision. Indeed, our approach would integrate well with longer wavelength electro-optic comb systems without requiring a second comb generator. Looking forward, an improved version could also provide a wider spectral span, if the comb generator is operated with zero carrier-envelope offset frequency, see discussion under (3), which is however beyond the scope of this work. We have added a corresponding discussion to the manuscript to clarify this point (cf. point (3)).

Second, there is a high interest in the ultra-violet spectral range that derives in part from low-redshift cosmological targets, where however no calibration has been demonstrated so far. Our work demonstrates spectrograph calibration at record-short wavelength and as such, in our view, clearly goes beyond the current state of the art in RV measurements in terms of wavelength. In addition, we note that electro-optic combs can readily be tuned in optical frequency across more than one line spacing. Our work thus also enables continuous determination of the instrument line-spread function in the UV. Besides the wavelength calibration itself, the characterization of the instrument line-spread function is key for astronomical precision measurements. To better communicate this point, we made revision to the introductory part of the manuscript, highlighting the importance of UV measurements. We also added in this context a new reference (Martins et al., “Cosmology and fundamental physics with the ELT-ANDES spectrograph”, arXiv:2311.16274, (2023)).

We agree with the Reviewer that a number of hard engineering challenges still lie ahead, along with critical advances in telescopes, instruments, astrophysics and data treatment, before the impact on astronomical observation can materialize. By demonstrating and validating the concept, we provide the basis on which further research and engineering efforts can be undertaken. To provide better context we have clarified in the manuscript that the comb is part of a larger community effort. Moreover, we added a detailed discussion on the technical challenges to the manuscript as further detailed under point (3).

(2) I have a hard time assessing the calibration precision/accuracy from the data of Fig. 4. The data of the drift appear to be inconclusive as to whether the comb is better than the FP—in fact the authors point out that the comb calibration is slightly noisier. And the measurements of the ThAr lines have uncertainty of 10’s of m/s that does not push to the state of the art. The fact that ESPRESSO has similar offsets does not build confidence in the calibration, but rather implies a deeper systematic issue independent of the comb. In all of this, the demo precision remains well above the state of the art for EPRV and at least 10-100x above the ultimate need of the EPRV community. That leaves me asking what significant advance is offered by this work if the nonlinear optics is already know and the RV precision is lacking?

We welcome these important questions by the Reviewer concerning the interpretation and significance of the data shown in Figure 4 and the significance of the advancement offered by our work. We will discuss each point in detail below:

The fundamental advance presented in our work is the generation of high-repetition rate ($f_{\text{rep}} \geq 18$ GHz) frequency combs in the ultra-violet spectral range directly from a high-repetition rate source at (infrared) telecom wavelength and the use of the generated UV combs for astronomical spectrograph calibration. This is the first demonstration of a comb-based calibration of a spectrograph in the important UV wavelength range, by definition advancing the state-of-the-art of spectrograph calibration. We emphasize, that in this first proof-of-concept demonstration, we do not aim at reaching the same level of RV precision that has been demonstrated at significantly more established longer wavelengths (a more detailed discussion on RV precision is provided below).

To reach our goal of UV calibration, we demonstrate harmonically generated combs in nanophotonic waveguides as a powerful concept that can directly connect mature high-repetition rate telecom wavelength lasers to their astrocomb application in the UV. We demonstrate this for an electro-optic comb generator, as well as conceptionally (without stabilization) for a microresonator source. Importantly, neither source relies on the suppression of unwanted comb lines that could get re-amplified in the supercontinuum process. This is to our knowledge the first time a precision metro-

logical experiment has been performed in the ultra-violet where individual lines can be accessed and used for spectrograph calibration. It also represents the first time a microresonator comb has been transferred to such short wavelength. Moreover, the underlying nonlinear dynamics of the spectral generation, including noise, have remained largely unexplored. In our revised manuscript we have made substantial addition to the manuscript and the new Supplementary Information (SI) document that discusses in detail the distinct physics (including noise, linewidth, continuum background) of harmonically generated $\chi^{(2)}$ -based astrocombs. To avoid redundant text, we refer the Reviewer to the discussion under point (6), where we provide more detail.

Here, we focus on discussing the data in Figure 4. The purpose of these data is to show key characteristics of the comb based calibrator, i.e., precision and accuracy, within the capabilities of the SOPHIE spectrograph:

- The drift measurement in Figure 4a shows that we can detect spectrograph drifts, including in the UV spectral range. Specifically, the data in Figure 4a show that the drift measured by our comb agrees with the FP-based drift measurement; as opposed to the FP cavity, which lacks a broadband UV illumination source, the comb also provide calibration in the UV. The noise of the comb-based calibration is slightly higher when compared to the FP-based calibration. This is expected from the narrower spacing of the comb lines, compared to the optimal spacing of the FP lines. As neighboring comb lines are not fully separated by the SOPHIE spectrograph, the line fitting procedure becomes more challenging resulting in a slightly elevated noise. However, this is not a fundamental limitation and simply caused by the relatively (in comparison to upcoming instruments) low resolving power of the spectrograph and the components that are currently available to us in our setup, which do not allow us to operate at frequencies higher than 18 GHz (higher frequency components are commercially available on the market). For clarity, we have expanded the description of the line fitting algorithm in the Methods section and also highlighted the challenge of the line fitting in the manuscript. In our reply under point (3), we will provide a continued discussion and explain why the RV precision is mainly limited by the spectrograph and not by the comb source.
- The measurement of ThAr emission line frequencies in Figure 4b shows that the comb can provide an accurate calibration, without relying on the ThAr standard; in other words, we can use the comb calibrated spectrograph to measure the ThAr lines and achieve agreement with the cataloged values within the precision of the measurement. Therefore, we conclude that our comb calibrator is accurate. In this measurement, the precision is limited by the number of ThAr lines, *not* by the frequency comb. Similar to a radial-velocity measurement, the precision is obtained statistically through a large number of spectral features. The presence of only ~ 100 ThAr lines (as opposed to several thousands of comb lines) is the reason why the precision of the measurement is limited. To avoid confusion in this point, we added a clarifying statement to the manuscript. Ideally, one would compare two independent UV frequency combs on a higher resolution spectrograph, however, this is beyond the scope of this present work.

Regarding the scatter of the measured ThAr frequencies in Figure 4b, we find them to be at the level of ± 25 m/s, which is at the expected level for the SOPHIE spectrograph. For comparison, the much more advanced ESPRESSO instrument achieves a scatter of the thorium lines with regard to the wavelength solution of ± 10 m/s. We added an explanation as well as a supporting reference (Schmidt et al., *Astronomy & Astrophysics* 646, A144 (2021)) to the manuscript.

Finally, while our measurement is compatible with a null results (i.e. agreement between comb and ThAr catalog), it is nevertheless interesting to note the deviation from zero is not observed for the first time and an offset from -10 to -15 meters has been reported based on a similar

comparison on ESPRESSO spectrograph (Schmidt et al., *Astronomy & Astrophysics* 646, A144 (2021)); although interesting, an investigation of this matter is not relevant to the present work.

In sum, the measurements in Figure 4 validate the key properties of a frequency comb calibrator, providing calibration for the first time in the UV. In this proof-of-concept demonstration we do not yet reach the precision of systems in less challenging wavelength ranges. However, considering that radial-velocity precision improves with spectral span and that this demonstration only utilized a narrow spectral interval for calibration, considering that our available comb line spacing was, for technical reasons, below the optimal spacing and that the SOPHIE instrument is not comparable to extreme-precision radial velocity instruments, we find that our first results bode well for future precision measurements that leverage the UV in addition to the more established wavelength ranges of astrocombs.

(3) One issue is that the repetition rate is not high enough for the spectrograph to fully resolve the comb lines. This makes it unclear if the sub-optimal calibration results arise from the comb or the spectrograph. Only discrete bands of just a few tens of nanometers are available for calibration, leaving a wide swath of spectrum un-covered. Finally, the line intensities are not flattened or stabilized in intensity. The community knows this is a significant issue in frequency comb calibration.

We thank the Reviewer for this comment regarding multi repetition rate, limited spectral span, and equalization/stabilization of line intensities, each of which we address below:

We agree with the Reviewer that the repetition rate of our comb generator was below that that would be optimal for the SOPHIE spectrograph. As we briefly pointed out above, the reason for that is that, our microwave source and our electro-optic modulators do not support higher modulation frequencies. This is a limitation of our specific components and not a general limitation. The impact of the low repetition rate is that adjacent comb lines are not fully separated by the spectrograph. Although the comb lines are narrow, once convolved with the instrument line-spread function of SOPHIE, they overlap. This complicates the line fitting procedure as neighboring comb lines and the underlying background flux have to be modeled for each line fit (as described in the Methods section). Based on this complexity, we do expect a penalty in the precision that manifest itself in a larger scatter in consecutive RV measurements, as in Figure 4a. To assess the magnitude of this effect, we can compare it to the best case scenario, i.e., that of the SOPHIE Fabry-Pérot calibrator with *ideal* spacing when considering immediately consecutive measurements. For such immediately consecutive measurements the disadvantage of the Fabry-Perot in terms stability and accuracy, when compared to a comb, are not relevant. Even for the dedicated Fabry-Perot calibrator the scatter in Figure 4a is only marginally smaller. Therefore, we conclude that the RV precision is limited by the spectrograph and not by the comb source. We revised the manuscript to add this previously missing clarification.

We agree with the Reviewer that our demonstration uses limited spectral span. Indeed, the main aim of our demonstration is to explore the new pathway to UV spectrograph calibration and achieving large spectral span was, at this stage, not a primary goal. While for optimized input pulses our system can generate significantly wider span (as Figure 2, and for a 18 GHz comb, the new Section 5 in the SI), several reasons motivated utilizing a narrower spectral span for this initial exploitative demonstration:

First, when the system is operated with larger spectral span (of each harmonic), the different harmonics will start to overlap. As follows from Eq. (1), an overall consistent frequency comb will only be formed if $\nu_0 = lf_{\text{rep}}$ with $l \in \mathbb{Z}$. This corresponds to arranging the driving electro-optic

oscillator to provide pulses with zero carrier-envelope offset frequency. We anticipate that this can be implemented either by a suitable offset lock to an auxiliary frequency comb as described by Obrzud et al. (“Broadband near-infrared astronomical spectrometer calibration and on-sky validation with an electro-optic laser frequency comb”, *Optics Express* 26, 26, 34830, 2018) or via techniques known from frequency comb metrology, e.g. Okubo et al. (“Offset-free optical frequency comb self-referencing with an f-2f interferometer”, *Optica* 5, 2, 288 (2018)), provided the harmonics show sufficient overlap. Implementing this is, however, beyond the scope of this initial demonstration.

Second, different from conventional $\chi^{(3)}$ -based comb generation, the linewidth close to the center of each harmonic is narrow and the background flux low (we discuss this in detail in our reply to (6)). However, both linewidth and background will grow with spectral distance from the center of each harmonic as a result of microwave phase noise in the electro-optic comb. The underlying mechanism is analogous to $\chi^{(3)}$ -based combs where linewidth and background grow with distance from the fundamental (i.e. 0th harmonic) pump laser. As in our current demonstration we do not utilize a phase-noise filter cavity we restrict our comb generation to a narrow spectral range where the background flux remains below the level where we would expect loss of line contrast (as seen by the spectrograph). Filtering cavities for phase noise, *not* suppression of unwanted comb lines (!), have been demonstrated as effective means of avoiding background flux due to microwave phase noise in electro-optic combs Beha et al. (“Electronic synthesis of light”, *Optica* 4, 4, 406, 2017), and we do not expect this to be a major obstacle when aiming at larger bandwidth in a follow-up system.

To clarify the potential for wider spectral span, we have added a corresponding discussion to the manuscript, highlighting the need for operating the initial comb source with $\nu_0 = lf_{\text{rep}}$ and a microwave phase-noise suppression filter. The three references mention above are in the manuscript. As mentioned already above, even without any further improvements, our approach could be combined with conventional infrared-based comb generators that can operate in the visible wavelength range but cannot currently access shorter blue and UV wavelength. Here, our waveguides could simply act as a wavelength extender.

Further, the Reviewer is correct that no line flattening or power equalization was attempted in our work. Moving towards higher RV precision, this is clearly one of the next steps. Fortunately, established methods, such as those based on spatial light modulators, proven at longer wavelength, can be transferred to the UV where the same optical components are also available commercially. To address this point, we now mention in the conclusion spectral flattening as a next step and also the availability of commercially available components for that task. We also added a corresponding reference Probst et al. (Proceedings of the SPIE, Volume 9147, doi:10.1117/12.2055784, 2014)

(4) As this was a proof-of-concept test on a spectrograph it might be argued that these issues are technical in nature and can be overcome. But this reader is left wondering about what fundamental advance this paper has enabled, particularly since supercontinuum in thin-film lithium niobate has been previously demonstrated. Beyond that, there remains a significant gap between proof of principle and the requirements of an actual frequency comb that astronomers can use. The results of this paper do not let us know if the engineering issues can be overcome such that the laser comb could be better than a FP, a Th-Ar lamp, or the combination of the two.

We welcome the Reviewer’s question about the fundamental advance this paper has enabled, considering in particular previous work in thin-film lithium niobate. We also appreciate the comment about the gap between a proof-of-principle demonstration and a comb astronomers can use to improve upon current Fabry-Perot and hollow-cathode lamp calibrators. We discuss these points below:

The fundamental processes underlying traditional supercontinuum generation are well understood and this understanding is mostly based on decades of work in nonlinear fiber optics. In comparison to nonlinear fibers, nano-photonic waveguides have over the last decade dramatically expanded the opportunities for broadband spectra through increased effective nonlinearity. They also provided efficient access to $\chi^{(2)}$ -nonlinear processes in materials such as lithium niobate or aluminium nitride, for example. By opening new parameter regimes for nonlinear optics, there are also new challenges and unknowns that emerge.

Considering lithium niobate waveguides for instance: Generating a broadband spectrum alone, although impressive, (see e.g. Figure 2) does not imply that the resulting spectrum can generally be used for optical precision metrology. As we explained above in our reply to (3), the different harmonics will exhibit different offset frequencies. As result, if they merge, there will be several interleaved combs with a relative spacing that corresponds to a fraction of the repetition rate (specifically: a multiple of the carrier-envelope offset frequency). Many demonstrations have provided prove for such interleaved combs. While they simplify the detection of the carrier-envelope offset frequency, they destroy one of the comb's main characteristics, i.e. that of equidistant comb lines. Thus importantly, we show in our work that the key characteristics of a frequency comb, equidistant lines with accurately known frequencies, are achieved. To our knowledge for the first time, we utilize such a UV comb source for a demanding meteorological experiment of astronomical spectrograph calibration, where individual lines can be accessed and used for metrology. We revised our manuscript to clearly explain this point and to present the condition $\nu_0 = lf_{\text{rep}}$ (or spectral gaps) as a necessary criterion for astrocombs (cf. point (3)); we are not aware that this condition has been discussed elsewhere.

Moreover, the noise properties in $\chi^{(2)}$ -based broadband combs are not well understood. In our revised manuscript, we made major additions regarding the distinct noise properties in $\chi^{(2)}$ -based combs combining data from the spectrograph with a detailed physical model of linewidth and background of the comb lines. The details of these revision are discussed in our reply under point (6). Therefore, in our work we demonstrate not only the generation of a $\chi^{(2)}$ -based spectrum from a high-repetition rate source, but in addition, its non-trivial use in a precision meteorological application. We present novel insights into the underlying noise properties (i.e. comb lineshape, linewidth, and background), which are critical for future advances of this technology.

Taking a bird's eye view on the evolution of astrocombs, we note that a main challenge has always been translating what can readily be achieved at low repetition rate to systems with high-repetition rate so that the comb lines can be separated by a spectrograph. Challenges range from high-average power induced damage to suppression of unwanted comb lines. Indeed, while low-repetition rate laser systems routinely create frequency combs across the THz, infrared, visible and ultraviolet part of the spectrum, the generating of astrocombs has, now more than 15 years after they have been proposed as calibrators, remained an active area of research. The significant effort devoted to the development of astrocombs reflects there huge scientific potential. It also shows that the transition from a low-repetition rate system to a high-repetition rate calibrator is non-trivial. Thus, although there are previous works on supercontinua, including in poled thin-film lithium niobate, we believe that our work reports a significant and non-incremental advancement.

We concur with the general assessment by Reviewer that there remains a substantial gap between our proof-of-concept demonstration and the practical use in astronomy. Nevertheless, our work demonstrates key features that motivate further advances towards closing this gap; these include spectral flattening, noise suppression (see discussion below), and zero-carrier envelope offset operation

(see reply under (3)).

Regarding the comparison between Fabry-Pérot (with ThAr) and comb-based calibrators, one fundamental advantage of frequency comb is that they may be linked to a precision metrological frequency standard, including the SI second. These standards are far superior to ThAr and provide, by definition, the best accuracy. Thus, on long time scales, years or decades, or data collection with different instruments, where potentially not even the same comb is used, the comb approach and its accuracy clearly outperforms any Fabry-Pérot cavity approach. In our case, ν_0 was linked to an optical clock transition in rubidium, and f_{rep} to a GPS-disciplined rubidium clock, both exceeding what can be achieved with ThAr by far.

Considering the short term RV precision, a comparison is more challenging as it depends not only on the calibrating light source but equally on the spectrograph and the data analysis. Fabry-Pérot etalons, due to their simplicity, are already fully mature devices that are operated routinely at observatories, while further research and development is needed to bring the inherently complex LFCs to the same technical readiness level. So far, to our knowledge, there has been no rigorous demonstration that a comb has outperformed a good Fabry-Pérot based calibration (in terms of short-term radial velocity precision). However, it is usually agreed on that ultimately the narrow lines of a comb offer several advantages over the much wider transmission features of Fabry-Pérot cavities. For instance, narrow comb lines may offer the measurement of the spectrograph's instrument line-spread function. They are also free-from imperfection in a Fabry-Pérot cavity that can result in irregular spectral resonance dispersion.

So overall, comb based calibrators are expected to ultimately outperform Fabry-Pérot based calibrators although this may only become apparent on long time-scales or when accuracy is key (which is the case for the science cases we considered). To clarify this point we have added a statement to the manuscript that explains the advantage of frequency combs as sources that can be referenced to precision metrological frequency standards.

Other clarifying points that the authors should address are:

(5) The authors need to strengthen the motivation. Astro spectroscopy at the cm/s level is a very complicated problem and the calibrator is just part of the challenge. Frequency comb calibration is necessary, but not the tall tent pole holding up the discovery of Earth 2.0. Similarly, can the authors provide better evidence that the calibration source is the limitation for the α dot measurements. Same for the Sandage test? What are the known requirements and limitations on the spectroscopy there? Right now, the state of the art in vis spectrographs is at the 20-30 cm/s level. Will the comb of the authors push this to significantly lower levels?

We appreciate the Reviewer's request to strengthen the motivation part of our manuscript and also agree with the Reviewer that the calibrator is only one part of a huge community challenge. For instance, achieving the extremely ambitious ANDES wavelength calibration goals will require substantial improvements over the current EPRV spectrographs and dedicated research in this field. This includes the development of new calibration sources, but also advanced methods and algorithms. A description of the goals for ANDES has recently been presented in Martins et al. 2023 10.48550/arXiv.2311.16274. One crucial aspect, highlighted multiple times by Martins et al. is coverage of the blue and UV spectral range down to 350 nm. Thus availability of astrocombs over the full wavelength range of ANDES (350 nm to 1800 nm), is needed for wavelength calibration itself but also to determine the spectral dependent instrument line-spread function; while the long wavelength part has been demonstrated multiple times, finding pathways towards shorter wavelengths has been an unresolved challenge. For instance, in the context of the redshift drift experiment, the crucial, nearly model-independent consistency check at $z \approx 2$ requires coverage of the UV around 365 nm.

To strengthen the motivation, we added a better description of this motivation to the introduction and also the reference (Martins et al.). The demonstrated experiment therefore marks an important step towards stable and accurate comb-based calibration that extends into short wavelength visible and ultraviolet spectral regions. This extension of the overall wavelength range where calibration can be performed will certainly be beneficial to improve RV precision. However, given the complexity of such precision experiments, and the ongoing developments in multiple areas, we feel a reliable estimate on the final RV precision cannot be made at this point.

Regarding specifically the investigation of a potential variability of the fine structure constant α we note that the classical approach in astronomy is to infer α from quasar absorption spectra and therefore at the place and time when the absorption happened, which can be up to 10 Gyr before the present time in the distant Universe. For clarity we note that this is for each measurement – at least in principle – a single-shot experiment and therefore does not measure a rate-of-change of α (although formally a rate may be inferred if one compares measurements at different cosmological redshifts or measurements in the distant Universe to the one on Earth). The measurement of the fine-structure constant requires an *accurate* wavelength calibration. In Schmidt et al. (2021, 10.1051/0004-6361/202039345, a thorough description of the current limitations is given. One of the key aspects where improvements are necessary is modeling of the instrumental line-spread function. Characterizing this wavelength-dependent function, however, requires a comb with narrow lines to reveal the line-spread function (Schmidt et al. in prep.). To improve the motivation part, we have added the importance of the determination of the line-spread function. In this context we also reference, in addition to Martins et al. (see above), Hirano et al. (“Precision radial velocity measurements by the forward-modeling technique in the near-infrared, arXiv:2007.11013, 2020) and Milaković/Jethwa (“A new method for instrumental profile reconstruction of high resolution spectrographs”, arXiv:2311.0524, 2023)

(6) What is the contribution of the continuum noise from the comb source itself? This is a known problem in astrocombs. From the experimental diagrams and description, it does not appear that there is a filter cavity for noise suppression, something that is common to broadband EO astrocombs generated through χ -3 processes. χ -2 broadening may have different noise processes than χ -3, but the question remains: is the low contrast truly due to the resolution of the spectrograph or is it indicative of noise washing out the comb lines? In particular, for the higher repetition rate microresonator example, this is not clear in the third harmonic data. The modes of Fig. 5 look well resolved, but there is a significant background flux between the comb teeth (does not go to zero). Can the authors explain this?

We thank the Reviewer for this important comment. To adequately address this comment, we have made major revisions and additions in our manuscript and in the new Supplementary Information (SI, Section 1), as we outline below:

The Reviewer correctly assumes that we did not use a cavity filter for noise suppression, which is necessary for very broadband combs based on $\chi^{(3)}$ processes. Indeed, if the UV comb would have been generated via a traditional $\chi^{(3)}$ -nonlinear processes and without a filter cavity, it would have been completely impossible to detect any comb line with the spectrograph. The weak non-zero fundamental background that is present in the fundamental near-infrared comb at $\sim 1.55 \mu\text{m}$ wavelength would have grown into a much larger background signal, completely drowning the comb line. In the case of a $\chi^{(2)}$ -based cascaded harmonic comb, the noise properties of the spectrum are however fundamentally different.

To appreciate the difference between $\chi^{(2)}$ - and $\chi^{(3)}$ based combs, we consider Eq. 1 in the main text. Both ν_0 and f_{rep} (Eq. 1, main text) exhibit non-zero phase noise described by the power spectral densities $S_{\phi}^{\nu_0}(f)$ and $S_{\phi}^{f_{\text{rep}}}(f)$, where f is the offset frequency from the respective carrier wave. The resulting phase noise of the comb lines is $S_{\phi}^{\nu_{m,n}} = m^2 S_{\phi}^{\nu_0} + n^2 S_{\phi}^{f_{\text{rep}}}$. The second term usually limits the useful span (i.e., a maximal $|n|$) of an electro-optic comb generator and noise-filtering must be applied if aiming at a wider range (Beha et al., 2017). Creating the comb spectra via cascaded $\chi^{(2)}$ -harmonics, however, permits reaching frequencies ν much higher than ν_0 while keeping $n \ll (\nu_{m,n} - \nu_0)/f_{\text{rep}}$. In other words, neglecting the small impact of $m^2 S_{\phi}^{\nu_0}$, the phase noise of comb lines of frequency $\nu_{m,n}$ that are part of the m^{th} harmonic comb does not grow in function of their frequency difference from ν_0 , but only with their distance from the m^{th} harmonics center frequency $\nu_{m,0}$. This permits generating UV astrocombs in a limited span around the harmonics' center frequencies $\nu_{m,0}$ from a near-infrared electro-optic comb generator without noise-filtering; wider spectra around the harmonics will still require noise filtering. The following Figure illustrates the difference between $\chi^{(2)}$ - and $\chi^{(3)}$ -based approaches.

Figure 1: **Phase noise in $\chi^{(2)}$ - and $\chi^{(3)}$ -based approaches.**

As we further detail in the Section 1 of the SI we can now estimate the lineshape of each comb line in the harmonic comb. It consists of a rather narrow peak on top of an almost flat background. Convolution of the computed lineshape with an (assumed) Gaussian SOPHIE instrument line-spread function allows us to determine the expected contrast for each comb line. Based on the understanding presented above, one would intuitively expect that the line contrast is highest in the center of each harmonics ($n = 0$) and then decreases the further the line is separated spectrally from the harmonic center frequency (i.e. $n^2 > 0$). Indeed, this is precisely what we observe, when we extract the contrast from the spectra shown in the main text Fig. 3c. The extracted contrast for both third and second harmonic of the comb is shown in the Supplementary Figure 2a,b. Assuming a certain repetition rate phase noise $S_{\phi}^{f_{\text{rep}}}$ (Supplementary Figure 2a,c) and width of the instrument line-spread function (see SI for details), we can well reproduce the observed line contrast values. To better illustrate the underlying comb lines shapes (prior to convolution), Supplementary Figure 2d shows their characteristic shapes for different values of n and Supplementary Figure 2e,f show the evolution the ratio between power in the narrow line vs power flat background. This new material, which we added in response to the Reviewer's comment, is a valuable addition to the manuscript and represents significant novel insights in the understanding of $\chi^{(2)}$ -based spectra.

The $\chi^{(2)}$ -based approach enables access to the UV wavelength range that, considering the center of each harmonic, is immune to the dramatic multiplication of phase noise one would observe if the same wavelength were generated in a $\chi^{(3)}$ process. However, there is nevertheless a growth of the phase noise for comb lines away from the center frequency of their harmonics, which for very broadband spectra around each harmonic would require the same noise filtering technique that has been established for $\chi^{(3)}$ -based combs. In our case, we restrict our comb generation to a narrow window around each harmonic where the background flux does not prevent line detection.

To provide additional information, also with regard to future implementation, we have added a detailed comparison of $\chi^{(2)}$ - and $\chi^{(3)}$ -based comb generation in Supplementary Figure 3. This figure shows in the left column (panels a,b) the conventional $\chi^{(3)}$ -based approach and in the right column (panels c,d) the approach based on $\chi^{(2)}$ processes. As a comparison between panels a and c shows, the width of the comb lines generated in the $\chi^{(2)}$ -based approach is, well below those generated in a $\chi^{(3)}$ -based approach (with noise suppression filter). Over the wavelength range utilized in our work, the $\chi^{(2)}$ -generated lines, even without noise suppression, are expected to be narrower than those one would expect from a $\chi^{(3)}$ -based approach with noise suppression filter. Panel b and c compare the relative power that is contained in the line and in the flat background. As can be seen, noise filtering can provide effective means of suppressing the background flux both for the $\chi^{(3)}$ -based approach as well as for extending the $\chi^{(2)}$ -based approach to wider bandwidth. As mention above, in our demonstration we restrict the wavelength interval of the harmonics so that the background flux remains low, even without the noise suppression cavity. For clarity, the operating interval in our work is indicated in panels c and d.

Considering the microresonator comb, we agree with the Reviewer that there is a background signal present in the spectra of Figure 5c in the main text, which reveals itself between the now more widely spaced lines (25 GHz as opposed to 18 GHz). A detailed discussion on noise in microresoantor combs would be beyond the scope of this work, however, in general one can expect that at large offset frequencies their noise is naturally suppressed (as they are high-finesse cavities). Although we have not studied this in detail, we believe that background originates not from repetition rate phase noise of the microresoantor itself but indeed from broadband amplified spontaneous emission (ASE) between the comb lines. Indeed, the sub-optimal coupling of the resonator (e.g. absence of a dedicated drop port) resulted in relatively weak output power with a strong residual continuous-wave pump line. To utilize this comb we removed the pump line via a bandpass filter that only retained a part of the comb. Overall, the output power was very low ($< 100 \mu\text{W}$), which in the >40 dB amplification process may likely have caused unwanted ASE. There are multiple approaches to reducing such ASE, including optimizing the microresonator, implementing a dedicated drop-port on the chip, optimizing the amplification stage, and finally, filtering the ASE noise optically after amplification similar to the approach that can be used for electro-optic combs.

In summary, to address the important comment by the Reviewer, we have added a new section in the Supplementary Material and substantially revised the manuscript with explanations along lines of the discussion provided above.

(7) Many of the leading spectrographs have combs that are capable of generating UV light. However, the real limitation is their lifetime and not the ability to resolve UV lines (that is simple grating optics). Photodarkening is the primary reason nonlinear fibers are a poor solution at short wavelengths. What can the authors say about this issue? It is unknown if lithium niobate has the same issue, and showing that it does not, especially for the intensities found within nanophotonic waveguides, would be a much stronger result.

We appreciate the comment regarding lifetime which is indeed an issue in conventional astro-combs that rely on spectral broadening in silica fiber.

Indeed, photo-induced damage is also a well-known in lithium niobate (e.g. Askin et al., Appl. Phys. Lett. 9, 72-74, 1966). Later studies (e.g. Bryan et al., Appl. Phys. Lett. 44, 847-849, 1984) found that magnesium-doping increases the damage threshold at 531 nm wavelength more than 100-fold to above 500 mW in a 40 μm laser spot size. However, in contrast to the significantly enhanced

damage threshold for green light, magnesium doping lowered this threshold for ultraviolet wavelength (Xu et al., Optics Lett. 25, 2, 2000). More recent work found that instead of magnesium-doping, doping with zirconium would provide improved damage resistance not only in the visible part of the spectrum but indeed also in the ultra-violet (Liu et al., Optics Lett. 35, 1, 10-12, 2010) tolerating intensities above 10^5 W/cm².

Our waveguide fabrication relies on commercial lithium niobate on insulator thin films. Currently, these are available either undoped or magnesium-doped. Based on the understanding of the photo-induced damage in lithium niobate presented above, we utilize in our case undoped lithium niobate thin films. While the power levels generated in our waveguides are by far sufficient for spectrograph calibration, the average integrated UV power levels are low and ≤ 100 μ W. Based on our experiments we have so far no indication of photo-induced damage in our waveguides.

We added a brief information about dopants and damage threshold to the main text, as well as, a statement on the absence of any noticeable photo-induced damage in our experiments.

As an additional experiment, we performed a dedicated test, where we coupled a blue laser diode to a lithium niobate waveguide (405 nm wavelength, 1-1.5 mW of coupled power). To perform this test, we fabricated dedicated lithium niobate waveguides that are not straight but for an S-bend on the waveguide such that input and output directions are offset from each other. This configuration enables accurate measurements of the coupled intensities without picking up stray light that has not propagated through the waveguide. Over 12 hours we delivered more than 60 J of ultraviolet power through the waveguide, which would correspond to more than one week of continuous operation in our experiment. Considering that astronomical spectrograph calibration is usually performed twice per day, requiring only approx. one hour of LFC operation this corresponds to 2 to 3 months of operation. Also in this experiment we did not see a degradation of the UV or infrared transmission through the waveguide. Although the waveguides were unpoled and would hence not have been suitable for spectrograph calibration, we also performed a supercontinuum generation experiment before and after UV irradiation resulting essentially in the same spectra. This new experiment along with a more detailed description is now presented in the SI, Section 4

Based on our experiments we believe that the photo-induced damage is not presently an issue. However, should photo-induced damage appear to be problematic, for instance in the context of broader spectra, zirconium doped lithium niobate could be utilized for waveguide fabrication (via custom doping if still not commercially available). Moreover, we note that in contrast to silica fibers, which are manufactured on one-by-one basis, hundreds of identical lithium niobate waveguides can be fabricated on one single chip. Mounting the chip on a translation stage (well established in fiber optics) it should be relatively straightforward to switch to a new waveguide before any photo-damage would be noticeable.

Overall, although photo-induced damage is a known effect in lithium niobate, we believe that the conditions and the non-observation of such damage in our experiments, the existence of higher damage threshold lithium niobate material and the ability to fabricate a large number of waveguides on one chip, provide a solid basis and a number of mitigation strategies (should they become necessary) in support of our approach.

(8) The core element of the TFLiN nanophotonic waveguide for UV generation has now been in place for several years. The authors point that out in several places. This includes the benefits and engineering that can be addressed with chirped periodic poling. The authors cite some of the recent papers, but similar designs with chirped poling were introduced in the following works. <https://doi.org/10.1364/NLO.2021.NTh1B.7> and https://doi.org/10.1364/CLEO_QELS.2022.FW4J.2. Also,

Ref 47 shows that the chirped poling the authors choose may not be optimal. Can you comment on the choice of poling that increases rather than decreases along the waveguide.

We agree with the Reviewer that one would initially assume that a poling from longer towards shorter periods would be a preferred design, where second, third and fourth harmonics are consecutively generated. We initially also pursued this idea.

In contrast to the ref. 47 mentioned by the Reviewer (now reference 51 and published in Nature Photonics), our design includes an initial poled section that provides immediate phase matching for the second harmonic not only half way along the waveguide but directly at the beginning. In our simulation (based on the open source pyChi code), we found that this can lead to significant spectral broadening (compare Fig. 2d and e) in comparison with an unpoled waveguide, lowering the required pulse energy. We attribute this to the relative strength of the $\chi^{(2)}$ -nonlinearity, where a second harmonic and the reverse process can provide a strong effective $\chi^{(3)}$ -like effect (e.g. Bruch et al. Nat. Photon., 15, 21-27 (2021), <https://doi.org/10.1038/s41566-020-00704-8>). In Figure 2f,g, we show a design where the initial SHG section is followed by a chirped poling. Of course the design of poling period would impact the resulting spectrum, but as long as suitable poling periods were present in the chirped poling section broadband UV/VIS light generation would happen. There was no clear advantage of one direction of chirp versus the other (We also confirmed this experimentally). For instance, in the design shown in Figure 2c the 4th harmonic signal grows first at shorter and then at longer wavelength and without previously enhancing the third harmonic. The complexity of the underlying dynamics including temporal synchronization of the relevant wavelength makes it in our view very difficult to come up with a simple design criterion and numeric simulation have proven useful. We have revised the manuscript to better describe the rational behind our waveguide design.

(9) Why is a range of pulse durations (50-150 fs) of the high rep rate pulse train incident on the chip? Factors of 3 in peak power are significant. Can the authors be more specific about what pulse durations are incident in the different situations.

We thank the Reviewer for this remark about the ambiguity and imprecision in the input pulse duration. In principle our 18 GHz electro-optic comb generator can produce pulses below 50 fs duration, supporting also the generation of rather broadband harmonics. To illustrate this, we have added an example spectrum in the SI, Section 5.

However, in our work we aim at the proof-of-principle of an astrocomb in the UV directly from a high-repetition rate source at telecom wavelength, and calibrating a spectrograph with it. To reduce the complexity in this proof-of-concept experiment we limited our comb to relatively narrow spans around the harmonics, so that noise suppression and ensuring consistent offsets between the harmonics would not be a concern. Thus, to limit the spectral span, we arranged the pulses from our electro-optic comb to be longer than their minimally possible duration. We estimate the pulse duration that was used in the presented experiment to be ca. 200 fs. We have specified the pulse duration in the manuscript and explained why longer pulses have been used.

(10) What is the FSR of the FP etalon? Are its modes well resolved? And how does its flux vary across the measured bands in comparison to the comb (which varies by up to 20 dB). Please report the difference in drift rates measured via the comb and the FP etalon? Are they statistically different?

We thank the Reviewer for pointing out the missing information on the FSR of the Fabry-Pérot etalon. The FSR of the Fabry-Pérot etalon is 38.9 GHz, resulting in well resolved and separated

modes. We added this information to the main text. The finesse should be relatively low, e.g. $\mathcal{F} \approx 12$, but is by itself not characterized in detail. The Fabry-Pérot is illuminated by a Energetic laser-driven light source (LDLS) EQ-99X. The number of counts per spectral line for Fabry-Pérot and LFC (optimized for either 3rd or 4th harmonic) are visualized in the following figure. The

Figure 2: **Counts, i.e. detected e^- , contained in each spectral line for third (green) and fourth harmonic (blue) of the LFC, as well as for each mode of the FP. The periodic modulation of the flux reflects the blaze function associated with each individual echelle order. Exposure times were 240 and 1 s for blue and green optimized LFC, respectively, and 180 s for the FP.**

spectrograph drifts measured either with the Fabry-Pérot or LFC are shown in Figure 4 of the main text. We have clarified the systematic drift rate as -2 m/s over 4 hours in the main text and that the different methods agree within the amplitude of their scatter (also see discussion under (2)).

(11) Please be more specific about how the resolving power of the spectrograph varies as a function of wavelength. And can that be correlated to the modulation depth of the lines as recorded by the spectrograph? In Fig. 3 it appears that modulation depth between 395 and 400 nm is less than that at wavelengths below 390 nm. I would have thought the resolving power of the spectrograph would be lower at the shortest wavelengths, making that the region of the spectrum with the lowest modulation depth. Similar point can be made in the spectra of the 3rd harmonic.

Indeed, the line-spread function of SOPHIE has not been measured and no such data is available through the instrument team. The resolving power of cross-dispersed echelle spectrographs is typically modulated along each individual spectral order, but is similar for neighboring orders. Therefore, there is no monotonic dependence on wavelength but a rather complex pattern that correlates with the structure of the echellogram. The spectra in Fig. 3b cover five spectral echelle orders in the UV. Therefore, non-trivial variations of the instrumental resolution across the presented spectral range (386 nm to 403 nm) are expected.

In Section 1 of the Supplementary Information (see discussion under point (6)), we estimated that the line-spread function has a full-width-at-half-maximum between 8 to 11 GHz in the green to UV wavelength ranges. This is in agreement with estimates based on the ThAr lines, which however by themselves are insufficient for a proper characterization of the instrumental line-spread function.

(12) What is the predicted calibration precision based on the photon noise and the number of comb lines employed? And how does that compare to the achieved precision and the design goals of the spectrograph itself. How does that compare for the comb, the FP interferometer and the Th-Ar

lamp? That would give information on the projected limitation of this present comb calibration.

We thank the Reviewer for this question. The calibration precision that could be expected in the best case from the present comb is given by the error bars in Figure 4a, which range from 10-30 cm/s. However, these error bars only take photon and detector read out noise into account and neglect other, more systematic sources of imprecision such as limitations of the spectrograph and determination of the comb line centers as discussed above. Indeed, the 1 m/s scatter of consecutive measurements in Figure 4a exceeds the error bars but is agreement with the SOPHIE design goals of 1 m/s internal calibration precision as outlined by Perruchot et al. (2008). We now point this out in the manuscript.

The goal of this proof-of-concept experiment is to demonstrate UV spectrograph calibration for the first time. Of high important for future application is that the comb source is directly derived from the most robust and mature laser technology available today, that in the telecom wavelength band around 1550 nm, and does not require suppression of unwanted comb lines. Moreover, access to the UV wavelength range is achieved efficiently through $\chi^{(2)}$ -nonlinear effects rather than $\chi^{(3)}$ -based effects, which would involve significantly higher power lasers and their associated challenges. Our revised manuscript presents a new approach and reveals novel insights into the underlying physical processes (including noise and its relation to lineshape). It provides a solid basis for future improvement of the new approach, including towards wider span and higher precision. We hope that in the light of our revision the Reviewer can recognizes our work, not only as interesting from an optics perspective, but also as a significant contribution to the development of astronomical calibration sources in an extremely challenging spectral domain.

REVIEWERS' COMMENTS

Reviewer #1 (Remarks to the Author):

The authors answered the questions with enough details and I thank them. I am satisfied with their answers.